# Wasserstein Logistic Regression with Mixed Features

**Aras Selvi**    **Mohammad Reza Belbasi**    **Martin B. Haugh**    **Wolfram Wiesemann**
Imperial College Business School, Imperial College London, United Kingdom
{a.selvi19, r.belbasi21, m.haugh, ww}@imperial.ac.uk

## Abstract

Recent work has leveraged the popular distributionally robust optimization paradigm to combat overfitting in classical logistic regression. While the resulting classification scheme displays a promising performance in numerical experiments, it is inherently limited to numerical features. In this paper, we show that distributionally robust logistic regression with mixed (*i.e.*, numerical and categorical) features, despite amounting to an optimization problem of exponential size, admits a polynomial-time solution scheme. We subsequently develop a practically efficient column-and-constraint approach that solves the problem as a sequence of polynomial-time solvable exponential conic programs. Our model retains many of the desirable theoretical features of previous works, but—in contrast to the literature—it does not admit an equivalent representation as a regularized logistic regression, that is, it represents a genuinely novel variant of logistic regression. We show that our method outperforms both the unregularized and the regularized logistic regression on categorical as well as mixed-feature benchmark instances.

## 1 Introduction

Consider a data set $(\boldsymbol{x}^i, y^i)_{i=1}^N$ with feature vectors $\boldsymbol{x}^i$ and associated binary labels $y^i \in \{-1, 1\}$. Classical logistic regression assumes that the labels depend probabilistically on the features via

$$\text{Prob}(y \mid \boldsymbol{x}) = \left[1 + \exp(-y \cdot [\beta_0 + \boldsymbol{\beta}^\top \boldsymbol{x}])\right]^{-1},$$

where the parameters $(\beta_0, \boldsymbol{\beta}) \in \mathbb{R}^{1+n}$ are estimated from the empirical risk minimization problem

$$\begin{aligned} \underset{(\beta_0, \boldsymbol{\beta})}{\text{minimize}} \quad & \frac{1}{N} \sum_{i=1}^N l_{\boldsymbol{\beta}}(\boldsymbol{x}^i, y^i) \\ \text{subject to} \quad & (\beta_0, \boldsymbol{\beta}) \in \mathbb{R}^{n+1} \end{aligned}$$

with the *log-loss* function $l_{\boldsymbol{\beta}}(\boldsymbol{x}, y) := \log\left(1 + \exp\left(-y \cdot [\beta_0 + \boldsymbol{\beta}^\top \boldsymbol{x}]\right)\right)$. Its compelling performance across many domains, the availability of mature and computationally efficient algorithms as well as its interpretability have all contributed to the widespread adoption of logistic regression [6, 18, 27].

Similar to other machine learning models, logistic regression is prone to overfitting, especially when the number of training samples is small relative to the number of considered features. Moreover, logistic regression can be sensitive to erroneous feature and label values as well as distribution shifts under which the training and test sets stem from different distributions. In recent years, distributionally robust (DR) optimization [3, 4] has been proposed to simultaneously address these challenges. To this end, the DR optimization paradigm models a machine learning task as a zero-sum game between the decision maker, who seeks to obtain the most accurate model (*e.g.*, the parameter vector $\boldsymbol{\beta}$ in a logistic regression), and a fictitious adversary who observes the decision maker's model and subsequently selects the worst data-generating distribution in the vicinity of the empirical distribution formed from the available training data. In this context, the similarity of distributions is commonly

36th Conference on Neural Information Processing Systems (NeurIPS 2022).

measured in terms of moment bounds [9, 39], $\phi$-divergences such as the Kullback-Leibler divergence [19, 22] or the Wasserstein distance [15, 25]. It has been observed that in many cases, the resulting DR machine learning models are equivalent to regularized versions of the same models, in which case distributionally robustness provides a new perspective on regularizers that are often selected *ad hoc* [21, 41]. Most importantly, DR machine learning models often admit computationally efficient reformulations as finite-dimensional convex optimization problems that can be solved in polynomial time with off-the-shelf solvers.

In this work we study a DR variant of the logistic regression where the distance between the empirical distribution and the unknown true data-generating distribution is measured by the popular Wasserstein (also known as Kantorovich-Rubinstein or earth mover's) distance [15, 25]. This problem has been first studied by [31], who show that the resulting DR logistic regression problem admits an equivalent reformulation as a polynomial-size convex optimization problem *if all features are numerical*. Applying similar techniques to the mixed-feature logistic regression problem, which appears to be most prevalent in practice, would result in an exponential-size convex optimization problem whose naïve solution with the methods of [31] does not scale to interesting problem sizes (*cf.* Section 4). Our contributions may be summarized as follows.

*(i)* On the *theoretical side*, we show that the complexity of DR mixed-feature regression crucially relies on the selected loss function. In particular, for the log-loss function employed in logistic regression, the problem—despite its natural representation as an exponential-size convex optimization problem—admits a polynomial-time solution scheme. We also show that in stark contrast to earlier variants of the problem, our mixed-feature regression does *not* admit an equivalent representation as a regularized problem. This provides compelling evidence that the DR logistic regression problem with mixed features is fundamentally different to the DR problem with only numerical features.

*(ii)* On the *computational side*, we propose a column-and-constraint scheme that solves the DR mixed-feature logistic regression as a sequence of polynomial-time solvable exponential conic programs. We show that the key step of our procedure, the identification of the most violated constraint, can be implemented efficiently for a broad range of metrics, despite its natural representation as a combinatorial optimization problem. Indeed, identifying the most violated constraint by brute force is out of the question, while the standard approach [43] of solving the most violated constraint problem would have an unacceptably high runtime.

*(iii)* On the *numerical side*, we show that our column-and-constraint scheme drastically reduces computation times over a naïve monolithic implementation of the regression problem. We also show that our model performs favorably on standard categorical and mixed-feature benchmark instances when compared against classical and regularized logistic regression.

The literature on DR machine learning under the Wasserstein distance is vast and rapidly growing. Recent works have considered, among others, the use of Wasserstein DR models in multi-label learning [14], generative adversarial networks [2, 8] and the generation of adversarial examples [40], density estimation [38] and learning Gaussian mixture models [20], graph-based semi-supervised learning [35], supervised dimensionality reduction [13] and reinforcement learning [1]. DR has also been found to help alleviate problems with overfitting [24, 21], label uncertainty [21] and distribution shifts [37, 36]. We refer to [21] for a recent review of the literature. Our work is most closely related to [31], and we compare our findings with the results of that work in Section 2.

We proceed as follows. Section 2 defines and analyzes the mixed-feature DR logistic regression problem, which is solved in Section 3 via column-and-constraint generation. We report numerical results in Section 4. Auxiliary material and all proofs are relegated to the appendix.

**Notation.** We define $\mathbb{B} = \{0, 1\}$ and $[N] = \{1, \dots, N\}$ for $N \in \mathbb{N}$. The set of all probability distributions supported on a set $\Xi$ is denoted by $\mathcal{P}_0(\Xi)$, while the Dirac distribution placing unit probability mass on $\boldsymbol{x} \in \mathbb{R}^n$ is denoted by $\delta_{\boldsymbol{x}} \in \mathcal{P}_0(\mathbb{R}^n)$. The indicator function $\mathbb{1}[\mathcal{E}]$ attains the value 1 (0) whenever the expression $\mathcal{E}$ is (not) satisfied. Finally, we use Roman d to denote differentials and to distinguish them from the $d$ used to denote distances.

## 2 Mixed-Feature DR Logistic Regression

Section 2.1 derives an exponential-size convex programming formulation of the DR logistic regression problem that serves as the basis of our analysis. Section 2.2 compares our formulation with a naïve model that treats categorical features as continuous ones. Section 2.3, finally, shows that our formulation, despite its exponential size, can be solved in polynomial time due to the benign structure of the log-loss function. It also establishes that in stark contrast to the literature, our formulation does not reduce to a regularized non-robust logistic regression, that is, it constitutes a genuinely novel variant of the logistic regression problem.

Our formulation enjoys strong finite-sample and asymptotic performance guarantees from the literature. Moreover, despite its exponential size, our model always accommodates worst-case distributions that exhibit a desirable sparsity pattern. We relegate these results to Appendix A.

### 2.1 Formulation as an Exponential-Size Convex Program

From now on, we consider a mixed-feature data set $\{\boldsymbol{\xi}^i := (\boldsymbol{x}^i, \boldsymbol{z}^i, y^i)\}_{i \in [N]}$ with $n$ numerical features $\boldsymbol{x}^i = (x_1^i, \ldots, x_n^i) \in \mathbb{R}^n$, $m$ categorical features $\boldsymbol{z}^i = (\boldsymbol{z}_1^i, \ldots, \boldsymbol{z}_m^i) \in \mathbb{C}(k_1) \times \ldots \times \mathbb{C}(k_m)$ and a binary label $y^i \in \{-1, +1\}$. Here, $\mathbb{C}(s) = \{\boldsymbol{z} \in \mathbb{B}^{s-1} : \sum_{j \in [s-1]} z_j \le 1\}$ for $s \in \mathbb{N} \setminus \{1\}$ represents the one-hot encoding of a categorical feature with $s$ possible values; in particular, $\mathbb{C}(2) = \mathbb{B}$ encodes a binary feature. We denote by $\mathbb{C} = \mathbb{C}(k_1) \times \ldots \times \mathbb{C}(k_m)$ and $\Xi = \mathbb{R}^n \times \mathbb{C} \times \{-1, +1\}$ the support of the categorical features as well as the data set, respectively, and we let $k = k_1 + \ldots + k_m - m$ be the number of slopes used for the categorical features.

If we had access to the true data-generating distribution $\mathbb{P}^0 \in \mathcal{P}_0(\Xi)$, we would solve the (non-robust) logistic regression problem

$$\begin{aligned}
\underset{\boldsymbol{\beta}}{\text{minimize}} \quad & \mathbb{E}_{\mathbb{P}^0}\left[l_{\boldsymbol{\beta}}(\boldsymbol{x}, \boldsymbol{z}, y)\right] \\
\text{subject to} \quad & \boldsymbol{\beta} = (\beta_0, \boldsymbol{\beta}_{\text{N}}, \boldsymbol{\beta}_{\text{C}}) \in \mathbb{R}^{1+n+k},
\end{aligned}$$

where $\mathbb{E}_{\mathbb{P}^0}$ denotes the expectation under $\mathbb{P}^0$, and where the log-loss function $l_{\boldsymbol{\beta}}$ now takes the form

$$l_{\boldsymbol{\beta}}(\boldsymbol{x}, \boldsymbol{z}, y) := \log\left(1 + \exp\left[-y \cdot \left(\beta_0 + \boldsymbol{\beta}_{\text{N}}^\top \boldsymbol{x} + \boldsymbol{\beta}_{\text{C}}^\top \boldsymbol{z}\right)\right]\right)$$

to account for the presence of categorical features. Since the distribution $\mathbb{P}^0$ is unknown in practice, the empirical risk minimization problem replaces $\mathbb{P}^0$ with the empirical distribution $\widehat{\mathbb{P}}_N := \frac{1}{N}\sum_{i=1}^N \delta_{\boldsymbol{\xi}^i}$ that places equal probability mass on all observations $\{\boldsymbol{\xi}^i\}_{i \in [N]}$. Standard arguments show that when these observations are i.i.d., the empirical risk minimization problem recovers the logistic regression under $\mathbb{P}^0$ as $N \longrightarrow \infty$. In practice, however, data tends to be scarce, and the empirical risk minimization problem exhibits an 'optimism bias' that is also known as overfitting [6, 18, 27], the error maximization effect of optimization [10, 23] or the optimizer's curse [34].

DR logistic regression combats the aforementioned overfitting phenomenon by solving the semi-infinite optimization problem

$$\begin{aligned}
\underset{\boldsymbol{\beta}}{\text{minimize}} \quad & \sup_{\mathbb{Q} \in \mathfrak{B}_\epsilon(\widehat{\mathbb{P}}_N)} \mathbb{E}_{\mathbb{Q}}\left[l_{\boldsymbol{\beta}}(\boldsymbol{x}, \boldsymbol{z}, y)\right] \\
\text{subject to} \quad & \boldsymbol{\beta} = (\beta_0, \boldsymbol{\beta}_{\text{N}}, \boldsymbol{\beta}_{\text{C}}) \in \mathbb{R}^{1+n+k},
\end{aligned} \tag{1}$$

where the *ambiguity set* $\mathfrak{B}_\epsilon(\widehat{\mathbb{P}}_N)$ contains all distributions $\mathbb{Q}$ in a (soon to be defined) vicinity of the empirical distribution $\widehat{\mathbb{P}}_N$, and where the expectation is taken with respect to $\mathbb{Q}$. Problem (1) can be interpreted as a zero-sum game between the decision maker, who chooses a logistic regression model parameterized by $\boldsymbol{\beta}$, and a fictitious adversary that observes $\boldsymbol{\beta}$ and subsequently chooses the 'worst' distribution (in terms of the incurred log-loss) from $\mathfrak{B}_\epsilon(\widehat{\mathbb{P}}_N)$. Contrary to the classical non-robust logistic regression, problem (1) guarantees to *over*estimate the log-loss incurred by $\boldsymbol{\beta}$ under the unknown true distribution $\mathbb{P}^0$ as long as $\mathbb{P}^0$ is contained in $\mathfrak{B}_\epsilon(\widehat{\mathbb{P}}_N)$. On the other hand, problem (1) recovers the empirical risk minimization problem when the ambiguity set $\mathfrak{B}_\epsilon(\widehat{\mathbb{P}}_N)$ approaches a singleton set that only contains the empirical distribution $\widehat{\mathbb{P}}_N$. Note that problem (1) is convex as the convexity of the logistic regression objective $\mathbb{E}_{\mathbb{Q}}\left[l_{\boldsymbol{\beta}}(\boldsymbol{x}, \boldsymbol{z}, y)\right]$ is preserved under the supremum operator. That said, the problem typically constitutes a semi-infinite program as it comprises finitely

many decision variables but—if the embedded supremum in the objective is brought to the constraints via an epigraph reformulation—infinitely many constraints whenever the ambiguity set harbors infinitely many distributions. As such, it is not obvious how the problem can be solved efficiently.

In this paper, we choose as ambiguity set the Wasserstein ball $\mathfrak{B}_\epsilon(\widehat{\mathbb{P}}_N) := \{\mathbb{Q} \in \mathcal{P}_0(\Xi) : W(\mathbb{Q}, \widehat{\mathbb{P}}_N) \leq \epsilon\}$ of radius $\epsilon > 0$ that is centered at the empirical distribution $\widehat{\mathbb{P}}_N$.

**Definition 1** (Wasserstein Distance). *The type-1* Wasserstein *(Kantorovich-Rubinstein, or* earth mover's*)* distance *between two distributions* $\mathbb{P} \in \mathcal{P}_0(\Xi)$ *and* $\mathbb{Q} \in \mathcal{P}_0(\Xi)$ *is defined as*

$$W(\mathbb{Q}, \mathbb{P}) := \inf_{\Pi \in \mathcal{P}_0(\Xi^2)} \left\{ \int_{\Xi^2} d(\boldsymbol{\xi}, \boldsymbol{\xi}') \, \Pi(\mathrm{d}\boldsymbol{\xi}, \mathrm{d}\boldsymbol{\xi}') \ : \ \Pi(\mathrm{d}\boldsymbol{\xi}, \Xi) = \mathbb{Q}(\mathrm{d}\boldsymbol{\xi}), \ \Pi(\Xi, \mathrm{d}\boldsymbol{\xi}') = \mathbb{P}(\mathrm{d}\boldsymbol{\xi}') \right\}, \quad (2)$$

*where* $\boldsymbol{\xi} = (\boldsymbol{x}, \boldsymbol{z}, y) \in \Xi$ *and* $\boldsymbol{\xi}' = (\boldsymbol{x}', \boldsymbol{z}', y') \in \Xi$*, while* $d(\boldsymbol{\xi}, \boldsymbol{\xi}')$ *is the ground metric on* $\Xi$*.*

The Wasserstein distance can be interpreted as the minimum cost of moving $\mathbb{Q}$ to $\mathbb{P}$ when $d(\boldsymbol{\xi}, \boldsymbol{\xi}')$ is the cost of moving a unit mass from $\boldsymbol{\xi}$ to $\boldsymbol{\xi}'$. The Wasserstein radius $\epsilon$ thus imposes a budget on the transportation cost that the adversary can spend on perturbing the empirical distribution $\widehat{\mathbb{P}}_N$.

We next define the ground metric $d$ that we use throughout the paper.

**Definition 2** (Ground Metric). *We measure the distance between two data-points* $\boldsymbol{\xi} = (\boldsymbol{x}, \boldsymbol{z}, y) \in \Xi$ *and* $\boldsymbol{\xi}' = (\boldsymbol{x}', \boldsymbol{z}', y') \in \Xi$ *with* $\boldsymbol{z} = (\boldsymbol{z}_1, \ldots, \boldsymbol{z}_m)$ *and* $\boldsymbol{z}' = (\boldsymbol{z}'_1, \ldots, \boldsymbol{z}'_m)$ *as*

$$d(\boldsymbol{\xi}, \boldsymbol{\xi}') := \|\boldsymbol{x} - \boldsymbol{x}'\| + d_{\mathrm{C}}(\boldsymbol{z}, \boldsymbol{z}') + \kappa \cdot \mathbb{1}[y \neq y'], \quad (3a)$$

*where* $\|\cdot\|$ *is any rational norm on* $\mathbb{R}^n$*,* $\kappa > 0$ *and the metric* $d_{\mathrm{C}}$ *on* $\mathbb{C}$ *satisfies*

$$d_{\mathrm{C}}(\boldsymbol{z}, \boldsymbol{z}') := \left( \sum_{i \in [m]} \mathbb{1}[\boldsymbol{z}_i \neq \boldsymbol{z}'_i] \right)^{1/p} \quad \textit{for some } p > 0. \quad (3b)$$

Intuitively, the ground metric $d$ measures the distance of the numerical features $\boldsymbol{x}$ and $\boldsymbol{x}'$ by the norm distance $\|\boldsymbol{x} - \boldsymbol{x}'\|$, whereas the distance of the categorical features $\boldsymbol{z}$ and $\boldsymbol{z}'$ is measured by (a power of) the number of discrepancies between $\boldsymbol{z}$ and $\boldsymbol{z}'$. Likewise, any discrepancy between the labels $y$ and $y'$ is accounted for by a constant $\kappa$, which allows for scaling between the features and the labels.

We first extend the convex optimization model of [31] for the DR continuous-feature logistic regression to an exponential conic model that accommodates for categorical features.

**Theorem 1** (Exponential Conic Representation). *The DR logistic regression problem* (1) *admits the equivalent reformulation*

$$\begin{aligned}
&\underset{\boldsymbol{\beta}, \lambda, \boldsymbol{s}, \boldsymbol{u}^\pm, \boldsymbol{v}^\pm}{\text{minimize}} && \lambda\epsilon + \frac{1}{N} \sum_{i \in [N]} s_i \\
&\text{subject to} && u_{i,\boldsymbol{z}}^+ + v_{i,\boldsymbol{z}}^+ \leq 1 \\
& && \left.\begin{aligned}
& (u_{i,\boldsymbol{z}}^+, 1, -s_i - \lambda d_{\mathrm{C}}(\boldsymbol{z}, \boldsymbol{z}^i)) \in \mathcal{K}_{\exp} \\
& (v_{i,\boldsymbol{z}}^+, 1, -y^i \boldsymbol{\beta}_{\mathrm{N}}^\top \boldsymbol{x}^i - y^i \boldsymbol{\beta}_{\mathrm{C}}^\top \boldsymbol{z} - y^i \beta_0 - s_i - \lambda d_{\mathrm{C}}(\boldsymbol{z}, \boldsymbol{z}^i)) \in \mathcal{K}_{\exp}
\end{aligned}\right\} \forall i \in [N], \ \forall \boldsymbol{z} \in \mathbb{C} \\
& && u_{i,\boldsymbol{z}}^- + v_{i,\boldsymbol{z}}^- \leq 1 \\
& && \left.\begin{aligned}
& (u_{i,\boldsymbol{z}}^-, 1, -s_i - \lambda d_{\mathrm{C}}(\boldsymbol{z}, \boldsymbol{z}^i) - \lambda\kappa) \in \mathcal{K}_{\exp} \\
& (v_{i,\boldsymbol{z}}^-, 1, y^i \boldsymbol{\beta}_{\mathrm{N}}^\top \boldsymbol{x}^i + y^i \boldsymbol{\beta}_{\mathrm{C}}^\top \boldsymbol{z} + y^i \beta_0 - s_i - \lambda d_{\mathrm{C}}(\boldsymbol{z}, \boldsymbol{z}^i) - \lambda\kappa) \in \mathcal{K}_{\exp}
\end{aligned}\right\} \forall i \in [N], \ \forall \boldsymbol{z} \in \mathbb{C} \\
& && \|\boldsymbol{\beta}_{\mathrm{N}}\|_* \leq \lambda \\
& && \boldsymbol{\beta} = (\beta_0, \boldsymbol{\beta}_{\mathrm{N}}, \boldsymbol{\beta}_{\mathrm{C}}) \in \mathbb{R}^{1+n+k}, \ \lambda \geq 0, \ \boldsymbol{s} \in \mathbb{R}^N \\
& && (u_{i,\boldsymbol{z}}^+, v_{i,\boldsymbol{z}}^+, u_{i,\boldsymbol{z}}^-, v_{i,\boldsymbol{z}}^-) \in \mathbb{R}^4, \ i \in [N] \text{ and } \boldsymbol{z} \in \mathbb{C}
\end{aligned}$$

$$(4)$$

*as a finite-dimensional exponential conic program, where* $\mathcal{K}_{\exp}$ *denotes the* exponential cone

$$\mathcal{K}_{\exp} := \mathrm{cl}\left(\{(a, b, c) \ : \ a \geq b \cdot \exp(c/b), \ a > 0, \ b > 0\}\right) \subset \mathbb{R}^3.$$

The $s_i$'s and $\lambda$ that appear in (4) are dual variables that arise from dual of the inner (*i.e.*, $\sup$) problem in (1). For each $\boldsymbol{z} \in \mathbb{C}$ and $i \in [N]$, $(u_{i,\boldsymbol{z}}^+, v_{i,\boldsymbol{z}}^+, u_{i,\boldsymbol{z}}^-, v_{i,\boldsymbol{z}}^-)$ is a vector of auxiliary variables that we use to model softplus constraints that arise from our intermediate analysis of problem (1) (Appendix E).

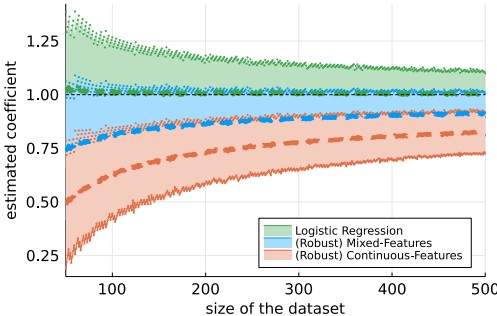 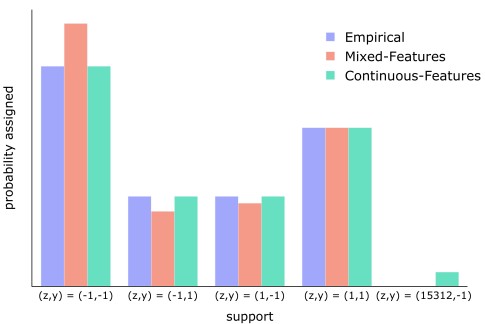

Figure 1: *Left:* Estimates of $\beta$ for the standard logistic regression, our mixed-feature as well as the continuous-feature model as a function of the size $N$ of the data set. All results are reported as averages over 2,000 statistically independent runs. *Right:* Comparison of the empirical distribution as well as the worst-case distributions for $\beta = 1$ under the two DR models for $N = 250$ samples. The probabilities are plotted on a log-scale to make small values visible.

Problem (4) is finite-dimensional and convex. Moreover, since exponential conic programs admit a self-concordant barrier function, they can be solved in polynomial time relative to their input size [28]. Due to the presence of categorical features in our regression problem, however, problem (4) comprises exponentially many variables and constraints, and a naïve solution of (4) would therefore require an *exponential* amount of time. The exponential size of problem (4) renders our formulation fundamentally different from the model of [31], and it significantly complicates the solution.

## 2.2 Comparison with Continuous-Feature DR Logistic Regression

Our formulation (4) of the mixed-feature DR logistic regression (1) accounts for categorical features $z \in \mathbb{C}$ at the expense of an exponential number of variables and constraints. It is therefore tempting to treat the categorical features as continuous ones and directly apply the continuous-feature-only reformulation of the DR logistic regression (1) proposed by [31]. In the following, we argue that such a reformulation would hedge against nonsensical worst-case distributions that in turn lead to overly conservative regression models. To see this, consider a stylized setting where the data set $(z^i, y^i)_{i \in [N]}$ comprises a single binary feature $z^i \in \{-1, +1\}$ that impacts the label $y^i \in \{-1, +1\}$ via the logistic model

$$\text{Prob}(y \mid z) = [1 + \exp(-y \cdot \beta z)]^{-1}$$

with $\beta = 1$. (While we use $z^i \in \{0, 1\}$ in the other sections of the paper, the analysis and results of this subsection and Appendix B assumed $z^i \in \{-1, +1\}$ as this made comparisons with [31] easier.) We attempt to recover this logistic model in two ways:

1. *Mixed-feature model.* We employ our DR logistic regression (1) with a single categorical feature, using $p = 1$ and $\kappa = 1$ in the ground metric (*cf.* Definition 2).

2. *Continuous-feature model.* We employ a variant of problem (1) that treats the categorical feature $z_i$ as a continuous one. We choose as ground metric

$$d(\boldsymbol{\xi}, \boldsymbol{\xi}') := \frac{1}{2}|z - z'| + \mathbb{1}[y \neq y'].$$

We therefore use the feature distances $\mathbb{1}[z \neq z']$ in the mixed-feature case and $\frac{1}{2}|z - z'|$ in the continuous-feature case. Note they both attain the value 0 (1) for equal (different) feature values.

Figure 1 (left) compares the mean values (dashed lines) as well as the 15% and 85% quantiles (shaded regions) of $\beta$ for the standard logistic regression, our mixed-feature as well as the continuous-feature model as a function of the size $N$ of the data set. For both DR models, we employed the same Wasserstein radius $\epsilon \propto 1/\sqrt{N}$, which is motivated by the finite sample guarantee presented in the next section (*cf.* Theorem 6). The figure shows that the continuous-feature model excessively shrinks its estimates of $\beta$, thus leading to overly conservative results. In fact, the true parameter value $\beta = 1$

consistently lies outside the confidence region of the continuous-feature model, independent of the sample size $N$. This is due to the fact that the continuous-feature model accounts for nonsensical worst-case distributions under which the categorical feature can take values outside its domain $\{-1, +1\}$, see Figure 1 (right). In fact, the continuous-feature model places non-zero probability on an extreme scenario under which the binary feature $z \in \{-1, +1\}$ attains the value $15,312$. (In Appendix B we explain why this value of $15,312$ arises.) Our mixed-feature model, on the other hand, restricts the worst-case distribution to the domain of the categorical feature and thus hedges against realistic distributions only.

We note that in practice, the radius $\epsilon$ of the Wasserstein ball will be chosen via cross-validation (*cf.* Section 4), in which case our mixed-feature model reliably outperforms the classical logistic regression on standard benchmark instances. While one may argue that the impact of nonsensical worst-case distributions in the continuous-feature model is alleviated by the cross-validated radii, Figure 1 offers strong theoretical and practical reasons against the use of a continuous-feature model for categorical or mixed-feature problems: The continuous-feature model hedges against a clumsily perturbed worst-case distribution that introduces a pronounced bias in the estimation. We also note that another possible approach would be to use the continuous-feature model but to restrict the support of binary features to $[-1, 1]$. Unfortunately, for the log-loss function no tractable reformulation with support constraints appears to be known; see [32] . Moreover, Section 4 will show that the mixed-feature model, despite its exponential size, can be solved quickly and reliably with our novel column-and-constraint approach from Section 3.

## 2.3 Complexity Analysis

We first show that despite its exponential size, the DR logistic regression problem (1) admits a polynomial-time solution. Key to this perhaps surprising finding is the shape of the loss function: while problem (1) is strongly NP-hard (and thus unlikely to admit a polynomial-time solution scheme) for generic loss functions, it can be solved in polynomial time for the log-loss function $l_{\boldsymbol{\beta}}$ employed in logistic regression.

**Theorem 2** (Complexity of the DR Logistic Regression (1))**.**

(i) *For generic loss functions $l_{\boldsymbol{\beta}}$, problem* (1) *is strongly NP-hard even if $n = 0$ and $N = 1$.*

(ii) *For the loss function $l_{\boldsymbol{\beta}}(\boldsymbol{x}, \boldsymbol{z}, y) = \log\left(1 + \exp\left[-y \cdot \left(\beta_0 + \boldsymbol{\beta}_{\mathrm{N}}^{\top}\boldsymbol{x} + \boldsymbol{\beta}_{\mathrm{C}}^{\top}\boldsymbol{z}\right)\right]\right)$ and the ground metric of Definition 2, problem* (1) *can be solved to $\delta$-accuracy in polynomial time.*

Recall that an optimization problem is solved to $\delta$-accuracy if a $\delta$-suboptimal solution is identified that satisfies all constraints modulo a violation of at most $\delta$. The consideration of $\delta$-accurate solutions is standard in the numerical solution of nonlinear programs where an optimal solution may be irrational.

A by now well-known result shows that when $m = 0$ (no categorical features), the DR logistic regression problem (1) reduces to a classical logistic regression with an additional regularization term $\|\boldsymbol{\beta_x}\|_*$ in the objective function when the output label weight $\kappa$ in Definition 2 approaches $\infty$ [31, 32]. We next show that this reduction to a classical regularized logistic regression no longer holds in our problem setting when categorical features are present.

**Theorem 3** (Absence of a Reformulation as a Regularized Problem)**.** *Even when the output label weight $\kappa$ approaches $\infty$ in the DR logistic regression* (1)*, problem* (1) *does not admit an equivalent reformulation*

$$
\begin{aligned}
&\underset{\boldsymbol{\beta}}{\text{minimize}} && \mathbb{E}_{\widehat{\mathbb{P}}_N}\left[l_{\boldsymbol{\beta}}(\boldsymbol{x}, \boldsymbol{z}, y)\right] + \mathfrak{R}(\boldsymbol{\beta}) \\
&\text{subject to} && \boldsymbol{\beta} = (\beta_0, \boldsymbol{\beta}_{\mathrm{N}}, \boldsymbol{\beta}_{\mathrm{C}}) \in \mathbb{R}^{1+n+k},
\end{aligned}
$$

*as a classical regularized logistic regression* for any regularizer $\mathfrak{R} : \mathbb{R}^{1+n+k} \to \mathbb{R}$.

Note that Theorem 3 does not only preclude the existence of a specific regularizer, but it excludes the existence of *any* regularizer, no matter how complex its dependence on $\boldsymbol{\beta}$ might be. We are not aware of any prior results of this form in the literature. Some insight for the result in Theorem 3 may be found via the special case we consider in its proof in Appendix E and in Remark 2 that follows it.

---

**Algorithm 1** Column-and-Constraint Generation Scheme for Problem (4).

---

Set $\text{LB}_0 = -\infty$ and $\text{UB}_0 = +\infty$
Choose (possibly empty) subsets $\mathcal{W}^+, \mathcal{W}^- \subseteq [N] \times \mathbb{C}$
**while** $\text{LB}_t \neq \text{UB}_t$ **do**
    Let $\theta^\star$ be the optimal value and $(\boldsymbol{\beta}, \lambda, \boldsymbol{s}, \boldsymbol{u}^\pm, \boldsymbol{v}^\pm)$ be a minimizer of problem (4) where:
- the first constraint set is only enforced for all $(i, \boldsymbol{z}) \in \mathcal{W}^+$;
- the second constraint set is only enforced for all $(i, \boldsymbol{z}) \in \mathcal{W}^-$;
- we only include those $(u_{i,\boldsymbol{z}}^+, v_{i,\boldsymbol{z}}^+)$ for which $(i, \boldsymbol{z}) \in \mathcal{W}^+$;
- we only include those $(u_{i,\boldsymbol{z}}^-, v_{i,\boldsymbol{z}}^-)$ for which $(i, \boldsymbol{z}) \in \mathcal{W}^-$.

    Identify a most violated index $(i^+, \boldsymbol{z}^+)$ of the first constraint set in (4) and add it to $\mathcal{W}^+$
    Identify a most violated index $(i^-, \boldsymbol{z}^-)$ of the second constraint set in (4) and add it to $\mathcal{W}^-$
    Let $\vartheta^+$ and $\vartheta^-$ be the violations of $(i^+, \boldsymbol{z}^+)$ and $(i^-, \boldsymbol{z}^-)$, respectively
    Update $\text{LB}_t = \theta^\star$ and $\text{UB}_t = \min\{\text{UB}_{t-1}, \ \theta^\star + \log(1 + \max\{\vartheta^+, \ \vartheta^-\})\}$
    Update $t = t + 1$
**end while**

---

**Algorithm 2** Identification of Most Violated Constraints in the Reduced Problem (4).

---

**for** $j \in \{1, \ldots, m\}$ **do**
    Find a feature value $\boldsymbol{z}_j^\star$ that minimizes $y^i \cdot \boldsymbol{\beta}_{\text{C},j}^\top \boldsymbol{z}_j$ across all $\boldsymbol{z}_j \in \mathbb{C}(k_j) \setminus \{\boldsymbol{z}_j^i\}$
**end for**
Let $\pi : [m] \to [m]$ be an ordering such that

$$y^i \cdot \boldsymbol{\beta}_{\text{C},\pi(j)}^\top (\boldsymbol{z}_{\pi(j)}^\star - \boldsymbol{z}_{\pi(j)}^i) \leq y^i \cdot \boldsymbol{\beta}_{\text{C},\pi(j')}^\top (\boldsymbol{z}_{\pi(j')}^\star - \boldsymbol{z}_{\pi(j')}^i) \qquad \forall 1 \leq j \leq j' \leq m$$

Set $\mathcal{W} = \emptyset$
**for** $\delta \in \{0, 1, \ldots, m\}$ **do**
    Update $\mathcal{W} = \mathcal{W} \cup \{\boldsymbol{z}\}$, where $\boldsymbol{z}_j = \boldsymbol{z}_j^\star$ if $\pi(j) \leq \delta$ and $\boldsymbol{z}_j = \boldsymbol{z}_j^i$ otherwise
**end for**
Determine the most violated constraint from the candidate set $\mathcal{W}$

---

## 3 Column-and-Constraint Solution Scheme

Our DR logistic regression problem (4) comprises exponentially many variables and constraints, which renders its solution as a monolithic exponential conic program challenging. Instead, Algorithm 1 employs a column-and-constraint generation scheme, *e.g.*, [43], that alternates between *(i)* the solution of relaxations of problem (4) that omit most of its variables and constraints and *(ii)* adding those variables and constraints that promise to maximally tighten the relaxations.

**Theorem 4.** *Algorithm 1 solves problem* (4) *in finitely many iterations. Moreover,* $\text{LB}_t$ *and* $\text{UB}_t$ *constitute monotone sequences of lower and upper bounds on the optimal value of problem* (4).

A key step in Algorithm 1 is the identification of most violated indices $(i, \boldsymbol{z}) \in [N] \times \mathbb{C}$ of the first and second constraint set in the reduced DR regression problem (4) for a fixed solution $(\boldsymbol{\beta}, \lambda, \boldsymbol{s}, \boldsymbol{u}^\pm, \boldsymbol{v}^\pm)$. For the first constraint set in (4), the identification of such indices requires for each data point $i \in [N]$ the solution of the combinatorial problem

$$\underset{\boldsymbol{z}}{\text{maximize}} \quad \min_{u^+, v^+} \left\{ u^+ + v^+ : \begin{bmatrix} (u^+, 1, -s_i - \lambda d_{\text{C}}(\boldsymbol{z}, \boldsymbol{z}^i)) \in \mathcal{K}_{\exp} \\ (v^+, 1, -y^i \boldsymbol{\beta}_{\text{N}}^\top \boldsymbol{x}^i - y^i \boldsymbol{\beta}_{\text{C}}^\top \boldsymbol{z} - y^i \beta_0 - s_i - \lambda d_{\text{C}}(\boldsymbol{z}, \boldsymbol{z}^i)) \in \mathcal{K}_{\exp} \end{bmatrix} \right\}$$

subject to $\quad \boldsymbol{z} \in \mathbb{C}$,

(5)

where an optimal value greater than 1 corresponds to a violated constraint; an analogous problem can be defined for the second constraint set in (4). Despite its combinatorial nature, the above problem can be solved efficiently by means of Algorithm 2.

**Theorem 5.** *Algorithm 2 identifies a most violated constraint for a given data-point* $i \in [N]$ *in time* $\mathcal{O}(k + n + m^2)$.

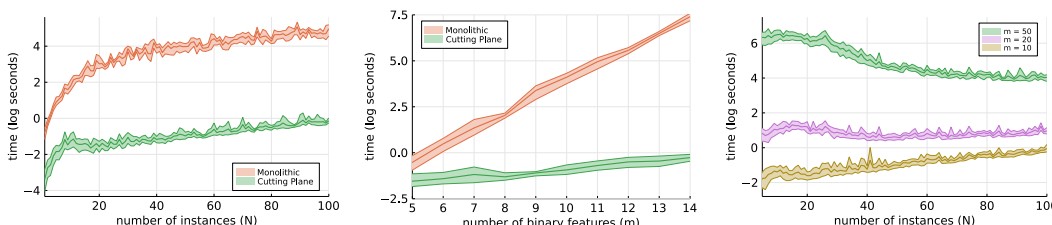

Figure 2: Runtime comparison between our column-and-constraint scheme and a naïve solution of problem (4) as a monolithic exponential conic program. *Left:* Runtimes for $m = 10$ binary features as a function of the number $N$ of data points. *Middle:* Runtimes for $N = 50$ as a function of $m$. *Right:* Runtimes of our column-and-constraint scheme only for varying combinations of $m$ and $N$. In all graphs, shaded regions correspond to 10%-90% confidence regions and bold lines report median values over 50 statistically independent runs. Note the log-scale in the plots.

Algorithm 2 first determines for each categorical feature $z_j$, $j = 1, \ldots, m$, the feature value $z_j^\star \in \mathbb{C}(k_j) \setminus \{z_j^i\}$ that, *ceteris paribus*, contributes to a maximal constraint violation in time $\mathcal{O}(k)$. The subsequent step sorts the different features $j = 1, \ldots, m$ in descending order of their contribution to constraint violations in time $\mathcal{O}(m \log m)$. The last step of Algorithm 2, finally, constructs the most violated constraint $z$ across those in which $\delta = 0, \ldots, m$ categorical feature values deviate from $z^i$ and subsequently picks the most violated constraint of these in time $\mathcal{O}(n + m^2)$. We note that the overall complexity of $\mathcal{O}(k + n + m^2)$ can be reduced to $\mathcal{O}(k + n + m \log m)$ by a clever use of data structures in the final step; for ease of exposition, we omit the details. Finally, since Algorithm 2 is applied to each data point $i \in [N]$, the overall complexity increases by a factor of $N$. We provide additional intuition behind Algorithm 2 immediately before the proof of Theorem 5 in Appendix E.

## 4 Numerical Results

Section 4.1 first compares the runtimes of our column-and-constraint scheme from Section 3 with those of solving the DR logistic regression problem (4) naïvely as a monolithic exponential conic program. We subsequently compare the classification performance of problem (1) with those of a classical unregularized and regularized logistic regression on standard benchmark instances with categorical features (Section 4.2) and mixed features (Appendix C).

All algorithms were implemented in Julia [5] (MIT license) and executed on Intel Xeon 2.66GHz processors with 8GB memory in single-core mode. We use MOSEK 9.3 [26] (commercial) to solve all exponential conic programs through JuMP [12] (MPL2 License). (We note the open source solvers Ipopt and CVXOpt could be used instead of MOSEK.) All source codes and detailed results are available on GitHub (`https://github.com/selvi-aras/WassersteinLR`).

### 4.1 Runtime Comparison with Monolithic Formulation

We first compare the computation times of our column-and-constraint scheme from Section 3 with those of solving the DR logistic regression problem (4) naïvely as a monolithic exponential conic program. To this end, we randomly generate synthetic logistic regression instances with varying numbers $N$ of data points and $m$ of binary features. While Figure 2 (left) shows that both approaches scale similarly in the number $N$ of data points, Figure 2 (middle) reveals that the solution of the monolithic formulation scales exponentially in the number $m$ of binary features. In contrast, our column-and-constraint scheme scales gracefully in both the number $N$ of data points and the number $m$ of binary features (*cf.* Figure 2, right). A similar behavior can be observed with more general, non-binary categorical features; we omit the results due to space constraints.

### 4.2 Performance on Categorical-Feature Instances

We next compare the classification performance (in terms of the out-of-sample classification error) of our unregularized ('DRO') and Lasso-regularized ('r-DRO') DR logistic regression problem (1) with those of a classical unregularized ('LR') as well as Lasso-regularized ('r-LR'), mass transportation-regularized (MT) [31, 32], and robust Wasserstein profile inference-regularized (PI) [7] logistic

| Data Set | $N$ | $k$ | $m$ | LR | DRO ($\kappa=1$) | DRO ($\kappa=m$) | r-LR | r-DRO ($\kappa=1$) | r-DRO ($\kappa=m$) | MT ($\kappa=1$) | MT ($\kappa=m$) | PI ($\alpha=0.05$) |
|---|---|---|---|---|---|---|---|---|---|---|---|---|
| breast-cancer | 277 | 42 | 9 | 29.56% | 29.15% | *28.55%*†‡ | 29.05% | 29.16% | 28.82% | 29.44% | 29.40% | *29.33%* |
| spect | 267 | 22 | 22 | 18.81% | 17.72% | *17.51%* | 17.79% | 18.38% | ***15.83%***†‡ | *18.49%* | 18.74% | 20.60% |
| monks-3 | 554 | 11 | 6 | ***2.07%*** | 2.08% | 2.10% | 2.14% | 2.19% | *2.13%* | *2.15%* | *2.15%* | 37.87% |
| tic-tac-toe | 958 | 18 | 9 | 1.92% | 1.69% | ***1.67%***† | *1.68%* | *1.68%* | 1.69% | 1.72% | 1.81% | 30.02% |
| kr-vs-kp | 3,196 | 37 | 36 | ***2.64%*** | 2.66% | ***2.64%*** | 2.69% | *2.66%* | *2.66%* | 2.77% | *2.74%* | 12.85% |
| balance-scale⋆ | 625 | 16 | 4 | 0.85% | 0.82% | *0.72%* | 0.73% | 0.71% | ***0.64%***†‡ | 0.72% | *0.69%* | 36.10% |
| hayes-roth⋆ | 160 | 11 | 4 | 17.00% | 16.31% | ***16.09%***†‡ | 18.66% | 18.19% | *17.78%* | 17.50% | *16.38%* | 37.59% |
| lymphography⋆ | 148 | 42 | 18 | 21.17% | 17.83% | ***16.72%***†‡ | 17.38% | 17.52% | *17.03%* | 19.41% | *17.83%* | 20.00% |
| car⋆ | 1,728 | 15 | 6 | ***4.73%*** | ***4.73%*** | ***4.73%*** | 4.99% | *4.76%* | 4.87% | ***4.73%*** | *4.77%* | 15.37% |
| splices⋆ | 3,189 | 229 | 60 | 6.80% | ***5.92%***†‡ | 6.66% | 6.22% | *5.99%* | 6.02% | 6.57% | 6.87% | 11.08% |
| house-votes-84 | 435 | 32 | 16 | 6.60% | *4.43%* | 5.37% | 4.54% | *4.41%* | 5.16% | *4.78%* | 5.54% | ***4.26%***† |
| hiv | 6,590 | 152 | 8 | 5.94% | 5.93% | 5.92% | ***5.90%***† | ***5.90%***† | ***5.90%***† | 6.04% | *6.03%* | 20.45% |
| primacy-tumor⋆ | 339 | 25 | 17 | 13.93% | ***13.66%*** | 13.76% | 14.51% | *14.06%* | 14.19% | 13.87% | *13.85%* | 19.18% |
| audiology⋆ | 226 | 92 | 69 | 14.18% | *14.03%* | 14.40% | 3.11% | ***2.80%***†‡ | 3.91% | 12.64% | *7.49%* | 15.49% |

Table 1: Classification errors of unregularized as well as Lasso-, mass transportation-, and profile inference-regularized variants of the classical logistic regression and our DR regression on UCI benchmark instances with categorical features only. The smallest error within each model group (unregularized vs. regularized) is highlighted in italics, whereas the smallest error overall (across all groups) is printed in bold. The dagger (†) and double dagger (‡) symbols next to the best model denote statistically significant improvements over LR and the second best model, respectively.

regression on the 14 most popular UCI data sets that only contain categorical features having more than 30 rows [11] (varying licenses). Instances with multiple output labels are indicated with a star; we convert them into instances with binary output labels by distinguishing between the majority class vs. all other classes. The instances vary in the number $N$ of data points, the number $m$ of categorical features as well as, accordingly, the number $k$ of slopes considered in the one-hot encoding of the categorical features (*cf.* Section 2.1). All results are reported as means over 100 random training set-test set splits (80%:20%). The radius $\epsilon \in \{0, 10^{-5}, \dots, 10^{-4}, \dots, 1\}$ of the Wasserstein ball as well as the Lasso penalty $\gamma \in \{0, \frac{1}{2} \cdot 10^{-5}, \dots, \frac{1}{2} \cdot 10^{-4}, \dots, \frac{1}{2}\}$ are selected via 5-fold cross-validation. We consider two variants of our DR logistic regression that employ a different output label weight ($\kappa = 1$ vs. $\kappa = m$) in the ground metric (*cf.* Definition 2). The results are reported in Table 1. (Run-times and are provided in Appendix D where we also describe how statistical significance was assessed.) The table shows that for the unregularized model, the classical logistic regression achieves the lowest classification error in 21% of the instances, whereas our DR logistic regression achieve the lowest classification error in 36% and 79% of the instances for $\kappa = 1$, $m$, respectively. (The double-counting of ties means these percentages do not sum to 100%.) For the regularized model, the results change to 14% (classical logistic regression) vs. 57% (each of our models). Within the numerical-feature Wasserstein DRO benchmarks, PI achieves the lowest classification error in 14% of the instances, whereas MT achieves in 42% and 57% of the instances for $\kappa = 1$, $m$, respectively. Globally, one of our methods is the winning approach in 12 out of 14 datasets (in the remaining 2 they are the second best approach without statistically significant inferiority). In 9 of the datasets our methods win strictly (without a tie), and in 7 of these datasets the improvements are statistically significant over all other approaches.

## 5   Conclusions

We proposed a new DR mixed-feature logistic regression model where the proximity between the empirical distribution and the unknown true data-generating distribution is measured by the popular Wasserstein distance. Despite its exponential-size formulation, we prove that the underlying optimization problem can be solved in polynomial time, and we develop a practically efficient column-and-constraint generation scheme for its solution. The promising performance of our model is demonstrated in numerical experiments on standard benchmark instances. We note that our column-and-constraint scheme readily extends to other DR mixed-feature machine learning models such as linear regression, support vector machines and decision trees. Applying our algorithm in those settings is a promising direction for future research. A further interesting direction is the development of tailored solution schemes rather than using off-the-shelf solvers for solving our DR mixed-feature logistic regression problem.

## Acknowledgments and Disclosure of Funding

Funding from the EPSRC grant EP/W003317/1 is gratefully acknowledged. The first author acknowledges support from The Alan Turing Institute. The authors are grateful for the comments and suggestions of the anonymous reviewers, which have helped to significantly improve the manuscript. This work has been produced without the involvement of any commercial entities that may cause a conflict of interest or competing interests.

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
