## Appendix A: Performance Guarantees

Wasserstein ambiguity sets benefit from measure concentration results that characterize the rate at which the empirical distribution $\widehat{\mathbb{P}}_N$ converges to the unknown true distribution $\mathbb{P}^0$. In the following, we review existing results from the literature to characterize the finite sample and asymptotic guarantees of our DR logistic regression (1).

**Theorem 6** (Finite Sample Guarantee). *Assume that $\mathbb{P}^0$ is light-tailed, that is, $\mathbb{E}_{\mathbb{P}^0}\left[\exp(\|\boldsymbol{\xi}\|^a)\right] \leq A$ for some $a > 1$ and $A > 0$. Then there are $c_1, c_2 > 0$ only depending on $\mathbb{P}^0$ through the light-tail parameters $a$, $A$ and the feature space dimensions $(n, m)$ such that any optimizer $\boldsymbol{\beta}^\star$ to (1) satisfies*

$$[\mathbb{P}^0]^N \left( \mathbb{E}_{\mathbb{P}^0}\left[l_{\boldsymbol{\beta}^\star}(\boldsymbol{x}, \boldsymbol{z}, y)\right] \leq \sup_{\mathbb{Q} \in \mathfrak{B}_\epsilon(\widehat{\mathbb{P}}_N)} \mathbb{E}_{\mathbb{Q}}\left[l_{\boldsymbol{\beta}}(\boldsymbol{x}, \boldsymbol{z}, y)\right] \right) \geq 1 - \eta$$

*for any confidence level $\eta \in (0, 1)$ and Wasserstein ball radius*

$$\epsilon \geq \left( \frac{\log(c_1/\eta)}{c_2 N} \right)^{1/\max\{m+n+1, 2\}} \cdot \mathbb{1}\left[ N \geq \frac{\log(c_1/\eta)}{c_2} \right] + \left( \frac{\log(c_1/\eta)}{c_2 N} \right)^{1/\alpha} \cdot \mathbb{1}\left[ N < \frac{\log(c_1/\eta)}{c_2} \right].$$

Recall that $[\mathbb{P}^0]^N$ in the statement of Theorem 6 refers to the $N$-fold product distribution of $\mathbb{P}^0$ that governs the data set $\{\boldsymbol{\xi}^i\}_{[i \in N]}$ upon which the optimizer(s) $\boldsymbol{\beta}^\star$ of problem (1) depend(s) via $\widehat{\mathbb{P}}_N$. Theorem 6 shows that with arbitrarily high probability $1 - \eta$, the optimal value $\sup_{\mathbb{Q} \in \mathfrak{B}_\epsilon(\widehat{\mathbb{P}}_N)} \mathbb{E}_{\mathbb{Q}}\left[l_{\boldsymbol{\beta}^\star}(\boldsymbol{x}, \boldsymbol{z}, y)\right]$ of our DR logistic regression (1) *over*estimates the loss $\mathbb{E}_{\mathbb{P}^0}\left[l_{\boldsymbol{\beta}^\star}(\boldsymbol{x}, \boldsymbol{z}, y)\right]$ incurred by any optimal solution $\boldsymbol{\beta}^\star$ under the unknown true distribution $\mathbb{P}_0$ as long as the radius $\epsilon$ of the Wasserstein ball $\mathfrak{B}_\epsilon(\widehat{\mathbb{P}}_N)$ is sufficiently large. Since the categorical features attain finitely many different values, the bound of Theorem 6 can be sharpened by replacing $m + n + 1$ with $n + 1$ if the constants $a$ and $A$ are adapted accordingly. We emphasize that the decay rate of $\mathcal{O}(N^{-1/(n+1)})$ in Theorem 6 is essentially optimal; see [21, §3].

To study the asymptotic consistency of problem (1) as well as the existence of sparse worst-case distributions, we first introduce a technical assumption.

**Definition 3** (Growth Condition). *We say that the DR logistic regression (1) satisfies the* growth condition *if (i) the hypotheses $\boldsymbol{\beta}$ are restricted to a bounded set $\mathcal{H} \subseteq \mathbb{R}^{1+n+k}$; and (ii) there is $\boldsymbol{\xi}^0 \in \Xi$ and $C > 0$ such that $l_{\boldsymbol{\beta}}(\boldsymbol{\xi}) \leq C[1 + d(\boldsymbol{\xi}, \boldsymbol{\xi}^0)]$ across all $\boldsymbol{\beta} \in \mathcal{H}$ and $\boldsymbol{\xi} \in \Xi$.*

**Lemma 1.** *If we restrict the hypotheses $\boldsymbol{\beta}$ to a bounded set $\mathcal{H} \subseteq \mathbb{R}^{1+n+k}$, then the DR logistic regression (1) satisfies the growth condition of Definition 3.*

We are now in the position to study the asymptotic consistency of problem (1).

**Theorem 7** (Asymptotic Consistency). *Under the assumptions of Theorem 6, we have*

$$\sup_{\mathbb{Q} \in \mathfrak{B}_{\epsilon_N}(\widehat{\mathbb{P}}_N)} \mathbb{E}_{\mathbb{Q}}\left[l_{\boldsymbol{\beta}^\star}(\boldsymbol{x}, \boldsymbol{z}, y)\right] \xrightarrow[N \to \infty]{} \mathbb{E}_{\mathbb{P}^0}\left[l_{\boldsymbol{\beta}^\star}(\boldsymbol{x}, \boldsymbol{z}, y)\right] \quad \mathbb{P}^0\text{-a.s.}$$

*whenever $(\eta_N, \epsilon_N)$ is set according to Theorem 6 for all $N \in \mathbb{N}$, $\sum_N \eta_N < \infty$, $\lim_{N \to \infty} \epsilon_N = 0$, and the growth condition in Definition 3 is satisfied.*

Theorem 7 shows that the DR logistic regression (1) achieves asymptotic consistency if the (un-)confidence parameter $\eta$ and the radius $\epsilon$ of the Wasserstein ball are reduced simultaneously. Thus, any optimal solution to (1) converges to the optimal solution of the (non-robust) logistic regression under the unknown true distribution $\mathbb{P}^0$ when the size of the data set increases.

The proof of Theorem 1 shows that the optimization problem characterizing the worst-case distribution $\mathbb{Q}^\star \in \mathfrak{B}_\epsilon(\widehat{\mathbb{P}}_N)$ comprises exponentially many decision variables. It is therefore natural to investigate the complexity of worst-case distributions to our DR logistic regression (1). The next result shows that there exist worst-case distributions that exhibit a desirable sparsity pattern: their numbers of atoms scale with the number of data samples.

**Theorem 8** (Existence of Sparse Worst-Case Distributions). *Assume that the growth condition in Definition 3 is satisfied. Then there are worst-case distributions $\mathbb{Q}^\star \in \mathfrak{B}_\epsilon(\widehat{\mathbb{P}}_N)$ satisfying*

$$\mathbb{E}_{\mathbb{Q}^\star}\left[l_{\beta^\star}(\boldsymbol{x}, \boldsymbol{z}, y)\right] = \sup_{\mathbb{Q} \in \mathfrak{B}_\epsilon(\widehat{\mathbb{P}}_N)} \mathbb{E}_{\mathbb{Q}}\left[l_{\beta^\star}(\boldsymbol{x}, \boldsymbol{z}, y)\right]$$

*such that $\mathbb{Q}^\star$ is supported on at most $N + 1$ atoms.*

Our performance guarantees in this section scale with the dimension of the feature space as we seek for a high confidence of the unknown true distribution being contained in our ambiguity set $\mathfrak{B}_\epsilon(\widehat{\mathbb{P}}_N)$. A dimension-independent performance guarantee can be obtained along the lines of [7] if one instead only seeks for a high confidence of the unknown true model $\beta^\star$ being contained in the union of optimal classifiers corresponding to the individual distributions $\mathbb{P}$ contained in $\mathfrak{B}_\epsilon(\widehat{\mathbb{P}}_N)$. We refer to [44] for a detailed review of the achievable performance guarantees of Wasserstein ambiguity sets with respect to the dimension of the feature space.

## Appendix B: Finding the Worst Distribution in Example of Section 2.2

We provide further details here on the derivation of the worst distribution associated with the benchmark in Section 2.2 where we treat the (only) binary feature as a numerical feature and use the DR logistic regression algorithm of [31]. We have a single binary feature $z \in \{-1, 1\}$ and there is no intercept term so the log-loss function is $l_\beta(z, y) = \log(1 + \exp(-\beta \cdot y \cdot z))$. The true unknown value of $\beta$ is $\beta = 1$, we have a data set $(z^i, y^i)_{i \in [N]}$, and we take $p = 1$, $\kappa = 1$ (distance metric parameters), and $\epsilon = 1/(2\sqrt{N})$. We would like to solve the worst distribution problem

$$\sup_{\mathbb{Q} \in \mathfrak{B}_\epsilon(\widehat{\mathbb{P}}_N)} \mathbb{E}_{\mathbb{Q}}\left[l_\beta(z, y)\right]. \tag{6}$$

We can do this by considering the following problem taken from [32, Thm 20] and adopted for our specific setting:

$$\begin{aligned}
\underset{\theta, \{\alpha_i\}_{i \in [N]}}{\text{maximize}} \quad & \theta + \frac{1}{N}\sum_{i=1}^{N}\left[(1 - \alpha_i)l_\beta(z^i, y^i) + \alpha_i l_\beta(z^i, -y^i)\right] \\
\text{subject to} \quad & \theta + \frac{1}{N}\sum_{i=1}^{N}\alpha_i = \epsilon - \gamma \\
& 0 \leq \alpha_i \leq 1, \quad i \in [N] \\
& \theta \geq 0.
\end{aligned} \tag{7}$$

It is parametrized by some $\gamma \in [0, \min\{\epsilon, 1\}]$ and [32, Thm 20] show that its optimal value for the case $\gamma = 0$ coincides with the optimal value of problem (6). Furthermore, if we denote by $(\theta^\star(\gamma), \{\alpha_i^\star(\gamma)\}_{i \in [N]})$ the optimal solutions to (7), then the sequence of probability distributions

$$\begin{aligned}
\mathbb{Q}_\gamma = \quad & \frac{1}{N}\sum_{i=2}^{N}\left[(1 - \alpha_i^\star(\gamma))\delta_{(z^i, y^i)} + \alpha_i^\star(\gamma)\delta_{(z^i, -y^i)}\right] + \frac{\eta(\gamma)}{N}\delta_{(z^1 + \frac{\theta^\star(\gamma)N}{\eta(\gamma)}, y^1)} \\
& + \frac{1 - \eta(\gamma)}{N}\left[(1 - \alpha_1^\star(\gamma))\delta_{(z^1, y^1)} + \alpha_1^\star(\gamma)\delta_{(z^1, -y^1)}\right]
\end{aligned} \tag{8}$$

where $\eta(\gamma) := \gamma/(\theta^\star(\gamma) + 2 - \epsilon + \gamma)$, constructs an asymptotically optimal solution to problem (6) as $\gamma \downarrow 0$. In order to explicitly characterize this sequence, we derive a closed-form solution to problem (7). Using the equality constraint to substitute for $\theta$ in the objective in (8) yields

$$\begin{aligned}
\underset{\{\alpha_i\}_{i \in [N]}}{\text{maximize}} \quad & \epsilon - \gamma - \frac{1}{N}\sum_{i=1}^{N}\alpha_i + \frac{1}{N}\sum_{i=1}^{N}\left[(1 - \alpha_i)l_\beta(z^i, y^i) + \alpha_i l_\beta(z^i, -y^i)\right] \\
\text{subject to} \quad & 0 \leq \alpha_i \leq 1, \quad i \in [N] \\
& \epsilon - \gamma \geq \frac{1}{N}\sum_{i=1}^{N}\alpha_i.
\end{aligned}$$

Ignoring the constant terms in the objective and re-arranging terms yields

$$\underset{\{\alpha_i\}_{i\in[N]}}{\text{maximize}} \quad \sum_{i=1}^{N} \alpha_i \left[ -1 + l_\beta(z^i, -y^i) - l_\beta(z^i, y^i) \right]$$
$$\text{subject to} \quad 0 \leq \alpha_i \leq 1, \quad i \in [N]$$
$$\epsilon - \gamma \geq \frac{1}{N} \sum_{i=1}^{N} \alpha_i.$$

Since $z^i \in \{-1, +1\}$, $i \in [N]$ holds, we have $l_\beta(z^i, -y^i) - l_\beta(z^i, y^i) \in \{-1, +1\}$, which implies that the coefficients of $\alpha_i$ in the objective function are non-positive. Hence, $\alpha_i^\star = 0$, $i \in [N]$ is an optimal solution to the problem. This implies $\alpha_i^\star(\gamma) = 0$, $i \in [N]$ and $\theta^\star(\gamma) = \epsilon - \gamma$ are optimal in problem (7).

We can then observe from (8) that the worst distribution places $1/N$ mass on each data point $i = 2, \ldots, N$, and the remaining $1/N$ mass is distributed as: $(1 - \eta(\gamma))/N$ mass on the data point $i = 1$ and $\eta(\gamma)/N$ mass on the point $(z^1 + \frac{(\epsilon-\gamma)N}{\gamma/2}, y^1)$ which is **not** in the data-set and in fact is an infeasible point. In our experiments, we take $\gamma = 10^{-3}$, because the optimal value of problem (7) numerically converges after this value, and one can then verify that in the setting we work with ($N = 250$) a mass of $\frac{10^{-3}}{500}$ is placed on the point with feature $z^1 + \frac{(\epsilon-\gamma)N}{\gamma/2} = 1 + \frac{(\frac{1}{1\sqrt{N}}-10^{-3})N}{10^{-3}/2} \approx 15,312$ and label $y^1$. This summarizes the specific approach we took to obtain the worst distribution in Figure 1 and demonstrates the key problem that arises with treating categorical variables as numerical.

## Appendix C: Numerical Results on UCI Data-Sets with Mixed Features

We repeat the experiment of the Section 4.2 for the five most popular mixed-feature instances of the UCI data set [11]. The results are reported in Table 2. All results are reported as medians over 20 random training set-test set splits (80%:20%). Cross-validation is applied for the same set of parameters as Section 4.2 over identical grids. The conclusions are qualitatively similar to those of Section 4.2. We highlight, however, that now only two of the lowest classification errors are achieved by one of the non-robust models, whereas the lowest classification error in each instance is obtained by at least one of the robust models we propose.

| Data Set | $N$ | $n$ | $k$ | $m$ | LR | DRO ($\kappa=1$) | DRO ($\kappa=m$) | r-LR | r-DRO ($\kappa=1$) | r-DRO ($\kappa=m$) | MT ($\kappa=1$) | MT ($\kappa=m$) | PI ($\alpha=0.05$) |
|---|---|---|---|---|---|---|---|---|---|---|---|---|---|
| credit-approval | 690 | 6 | 36 | 9 | 14.13% | 13.77% | **13.04%**†‡ | 14.13% | 14.13% | **13.04%**†‡ | 14.13% | 14.13% | *13.41%* |
| annealing⋆ | 798 | 6 | 46 | 32 | 2.52% | 2.83% | 2.52% | 2.20% | **1.89%**†‡ | **1.89%**†‡ | 2.52% | 2.20% | 12.58% |
| contraceptive⋆ | 1,473 | 2 | 15 | 7 | *33.16%* | *33.16%* | *33.16%* | **32.82%**† | 32.99% | **32.82%**† | *33.16%* | *33.16%* | 39.63% |
| hepatite | 155 | 6 | 23 | 13 | **16.13%** | **16.13%** | 19.35% | 19.35% | 17.74% | **16.13%** | **16.13%** | **16.13%** | 17.74% |
| cylinder-bands | 539 | 19 | 43 | 15 | 23.36% | **21.50%**†‡ | **21.50%**†‡ | 23.36% | **21.50%**†‡ | 22.43% | 22.43% | 22.43% | 30.84% |

Table 2: Classification errors of unregularized and Lasso-regularized variants of the classical logistic regression and our DR regression on UCI benchmark instances with mixed features. We use the same notation and highlighting conventions as in Table 1.

| Data Set | $N$ | $n$ | $k$ | $m$ | LR | DRO ($\kappa=1$) | DRO ($\kappa=m$) | r-LR | r-DRO ($\kappa=1$) | r-DRO ($\kappa=m$) | MT ($\kappa=1$) | MT ($\kappa=m$) | PI ($\alpha=0.05$) |
|---|---|---|---|---|---|---|---|---|---|---|---|---|---|
| credit-approval | 690 | 6 | 36 | 9 | 0.11 | 53.46 | 55.89 | 0.09 | 36.77 | 43.58 | 0.36 | 0.56 | 0.07 + 0.08 |
| annealing⋆ | 798 | 6 | 46 | 32 | 0.18 | 62.96 | 71.09 | 43.58 | 22.91 | 0.24 | 0.36 | 0.58 | 0.10 + 0.12 |
| contraceptive⋆ | 1,473 | 2 | 15 | 7 | 0.14 | 86.06 | 81.70 | 0.15 | 91.10 | 84.04 | 0.52 | 0.74 | 0.08 + 0.03 |
| hepatite | 155 | 6 | 23 | 13 | 0.05 | 9.01 | 10.59 | 0.02 | 2.49 | 4.86 | 0.06 | 0.10 | 0.01 + 0.05 |
| cylinder-bands | 539 | 19 | 43 | 15 | 0.13 | 96.10 | 106.48 | 0.09 | 67.09 | 41.63 | 0.32 | 0.59 | 0.06 + 0.12 |

Table 3: Median runtimes (in seconds) associated with Table 2.

## Appendix D: Further Details on Numerical Experiments

Throughout the numerical experiments, we fixed $p = 1$ in the ground metric (*cf.* Definition 2).

**Synthetic data sets:** In Section 4.1 we generate synthetic data sets with $N$ data points and $m$ binary features. The data generation process is summarized in Algorithm 3.

---

**Algorithm 3** Construction of synthetic data sets in Section 4.1.

---

Sample the components of $\beta_0$ and $\boldsymbol{\beta}_{\mathrm{C}}$ i.i.d. from a standard normal distribution
Normalize $\beta_0$ and $\boldsymbol{\beta}_{\mathrm{C}}$ by dividing them with $\|\boldsymbol{\beta}\|_2$, where $\boldsymbol{\beta} = (\beta_0, \boldsymbol{\beta}_{\mathrm{C}})$
**for** $i \in \{1, \ldots, N\}$ **do**
    Construct $\boldsymbol{z}^i \in \mathbb{C}(2) \times \ldots \times \mathbb{C}(2)$ by sampling $\boldsymbol{z}^i \sim \{0,1\}^m$ uniformly at random
    Find $p^i$, the probability of the $i$th data point having label $+1$, by using the 'true' $\beta_0$ and $\boldsymbol{\beta}_{\mathrm{C}}$:

$$p^i = \left[1 + \exp(-[\beta_0 + \boldsymbol{\beta}_{\mathrm{C}}^\top \boldsymbol{z}^i])\right]^{-1}$$

    Sample $y^i \in \{-1, 1\}$ from a Bernoulli distribution with parameter $p^i$
**end for**
The synthetic data set is the collection of data points $\{(\boldsymbol{z}^i, y^i)\}_{i \in [N]}$ constructed above

---

**UCI data sets:** The datasets of Sections 4.2 and Appendix C were taken from the UCI repository [11]. Missing values (NaNs) were encoded as a new category of the corresponding feature; the dataset breast-cancer is an exception, where rows with missing values were dropped. Some data sets include features that are derivatives of the labels, primary keys of the instances (*e.g.*, the auidology data set), or they have features with only one possible category (*e.g.*, the annealing and cylinder-bands data sets). We removed such features manually. We also removed rows that were readily identified as erroneous (*e.g.*, two rows in cylinder-bands had less columns than required). Data sets with multi-class (*i.e.*, non-binary) labels were converted to binary labels by distinguishing between the majority label class and all other classes. If a data set includes separate training and test sets, we merged them and subsequently applied our training set-test set split as described in the main paper. As per standard practice, we randomly permuted the rows of each data set before conducting the splits into training and test sets. For the detailed data processing steps, we refer to the GitHub repository accompanying this paper, which also contains a sample Python 3 script.[1]

**Implementation of the column-and-constraint generation scheme:** In our implementation, we switch between solving the primal and dual exponential conic reformulation of the DR logistic regression problem, as we found that sometimes the dual problem can be solved more easily than the primal. We have further implemented an 'easing' step in the column-and-constraint generation scheme that periodically deletes constraints from $\mathcal{W}^+$ and $\mathcal{W}^-$ whose slacks exceed a pre-specified threshold. To this end, note that the constraints $u_{i,\boldsymbol{z}}^+ + v_{i,\boldsymbol{z}}^+ \leq 1$, $(i, \boldsymbol{z}) \in \mathcal{W}^+$, in the relaxations of the exponential conic reformulation (4) have a slack ranging between $0$ and $1$ by construction. We implemented variants of our column-and-constraint generation scheme that conduct easing steps either every 200 iterations or in iteration $t = 100 \cdot 1.5^k$, $k \in \mathbb{N}$. We have also implemented variants that keep the slack threshold constant at $0.05$ and where this threshold starts at $0.02$ and is subsequently increased by $0.02$ in each easing step. In all of our variants, a constraint is deleted at most once, that is, it is no longer considered for deletion if it has been reintroduced after a prior deletion. This ensures that Algorithm 1 terminates in finite time without cycling. Analogous steps have been implemented for the constraints $u_{i,\boldsymbol{z}}^- + v_{i,\boldsymbol{z}}^- \leq 1$, $(i, \boldsymbol{z}) \in \mathcal{W}^-$.

**Determining statistical Significance:** In the numerical experiments of Section 4.2 we compare the means of 100 out-of-sample errors attained by several logistic regression methods and also identify which methods appeared to be statistically significant. In Table 1, for example, a dagger (†) symbol next to the winning approach denotes statistically significant error improvement over the standard logistic regression ('LR') and a double dagger (‡) symbol denotes such improvement over the second best approach. Note that if the winning approach is a variant of the DRO methods we propose, then the second best approach is taken over the methods excluding our methods for a fair comparison.

We add a dagger (†) to the winning approach according to the following approach. Firstly, we subtract (element-wise) the vector of errors attained by standard logistic regression from the vector of errors attained by the winning approach. Each element of the new vector is the 'additional error' the winning approach has compared to the standard logistic regression. We then try to reject the hypothesis (at $5\%$-significance level) that this additional error is non-negative (*i.e.*, we try to reject

---

[1] https://github.com/selvi-aras/WassersteinLR

| Data Set | $N$ | $k$ | $m$ | LR | DRO ($\kappa = 1$) | DRO ($\kappa = m$) | r-LR | r-DRO ($\kappa = 1$) | r-DRO ($\kappa = m$) | MT ($\kappa = 1$) | MT ($\kappa = m$) | PI ($\alpha = 0.05$) |
|---|---|---|---|---|---|---|---|---|---|---|---|---|
| breast-cancer | 277 | 42 | 9 | 0.06 | 218.61 | 214.83 | 0.07 | 15.19 | 20.52 | 0.08 | 0.09 | 0.02 + 0.07 |
| spect | 267 | 22 | 22 | 0.08 | 32.05 | 38.95 | 0.07 | 20.81 | 35.50 | 0.07 | 0.09 | 0.02 + 0.03 |
| monks-3 | 554 | 11 | 6 | 0.12 | 46.83 | 42.59 | 0.08 | 31.04 | 42.53 | 0.12 | 0.14 | 0.03 + 0.02 |
| tic-tac-toe | 958 | 18 | 9 | 0.20 | 71.26 | 69.80 | 0.13 | 61.88 | 64.33 | 0.23 | 0.27 | 0.06 + 0.04 |
| kr-vs-kp | 3,196 | 37 | 36 | 1.65 | 217.98 | 40.64 | 0.71 | 21.79 | 11.29 | 2.11 | 2.87 | 0.34 + 0.08 |
| balance-scale⋆ | 625 | 16 | 4 | 0.15 | 13.60 | 17.41 | 0.19 | 19.74 | 24.04 | 0.21 | 0.19 | 0.03 + 0.03 |
| hayes-roth⋆ | 160 | 11 | 4 | 0.06 | 3.67 | 3.92 | 0.02 | 1.12 | 0.85 | 0.06 | 0.06 | 0.01 + 0.02 |
| lymphography⋆ | 148 | 42 | 18 | 0.09 | 145.46 | 154.69 | 0.03 | 4.83 | 11.91 | 0.08 | 0.09 | 0.01 + 0.07 |
| car⋆ | 1,728 | 15 | 6 | 0.46 | 168.09 | 112.33 | 0.28 | 210.35 | 181.63 | 0.57 | 0.52 | 0.11 + 0.05 |
| splices⋆ | 3,189 | 229 | 60 | 4.25 | 4,246.36 | 4,393.97 | 1.48 | 1,169.80 | 453.73 | 6.13 | 10.21 | 1.55 + 1.77 |
| house-votes-84 | 435 | 32 | 16 | 0.18 | 88.56 | 105.66 | 0.06 | 21.6 | 20.71 | 0.12 | 0.16 | 0.03 + 0.08 |
| hiv | 6,590 | 152 | 8 | 8.44 | 2,874.62 | 1,598.83 | 2.20 | 2,119.26 | 768.01 | 10.94 | 12.93 | 2.18 + 1.14 |
| primacy-tumor⋆ | 339 | 25 | 17 | 0.22 | 61.75 | 57.81 | 0.06 | 11.93 | 13.21 | 0.08 | 0.12 | 0.02 + 0.04 |
| audiology⋆ | 226 | 92 | 69 | 0.05 | 18.52 | 16.10 | 0.04 | 9.19 | 5.28 | 0.11 | 0.13 | 0.04 + 0.16 |

Table 4: Mean runtimes (in seconds) associated with Table 1.

the hypothesis that the standard logistic regression is at least as good). To this end, we compute the t-statistic for the mean of the additional errors vector with a hypothesis mean of 0 and sample size of 100. We then compute the cumulative probability of this value via a one-sided t-test (with $100 - 1$ degrees of freedom) to obtain a p-value. If this value is less than 0.05, then we reject the hypothesis, concluding that the improvement is significant. Note that, with this approach we implicitly assume the out-of-sample errors are independent, which is typically not the case [29]. Hence, we acknowledge that these tests of significance are only approximate. The presence of a double dagger is determined analogously.

**Computing environment:** We implemented all algorithms in Julia using MOSEK's exponential cone solver as well as JuMP to interact with the solver. We used the high performance computing cluster of Imperial College London, which runs a Linux operating system as well as Portable Batch System (PBS) for scheduling the jobs. We ran our experiments as batch jobs on Intel Xeon 2.66GHz processors with 8GB memory in single-core and single-thread mode. The job descriptions as well as all PBS commands are included in the GitHub repository.

**Runtimes corresponding to Table 1:** Table 4 presents the mean runtimes of each method on every dataset for the experiments corresponding to Table 1. Here, the times reported for 'LR' and 'MT' are the times the solver took to solve the corresponding optimization problems. The times corresponding to 'DRO' methods we propose are the total solver times summed for each sub-problem solved during the column-and-constraint generation scheme, including the identification of the most violated constraints. The times corresponding to 'PI' display the solution time to solve the regularized logistic regression problems plus the time it takes to identify the regularization parameter as proposed in [7]. The columns of Table 4 are identical to those in Table 1.

**Error bars of the experiments:** Figures 3 and 4 report the error distributions corresponding to the experiments of Tables 1 and 2, respectively. In these figures, each of the six sub-plots reports the errors of a specific model (*e.g.*, regularized DRO with $\kappa = 1$). In each sub-plot, the horizontal axis lists the considered data sets, and the vertical axis visualizes the 100 test errors in box-and-whisker representation (where the boxes enclose the 25% and 75% quartiles).

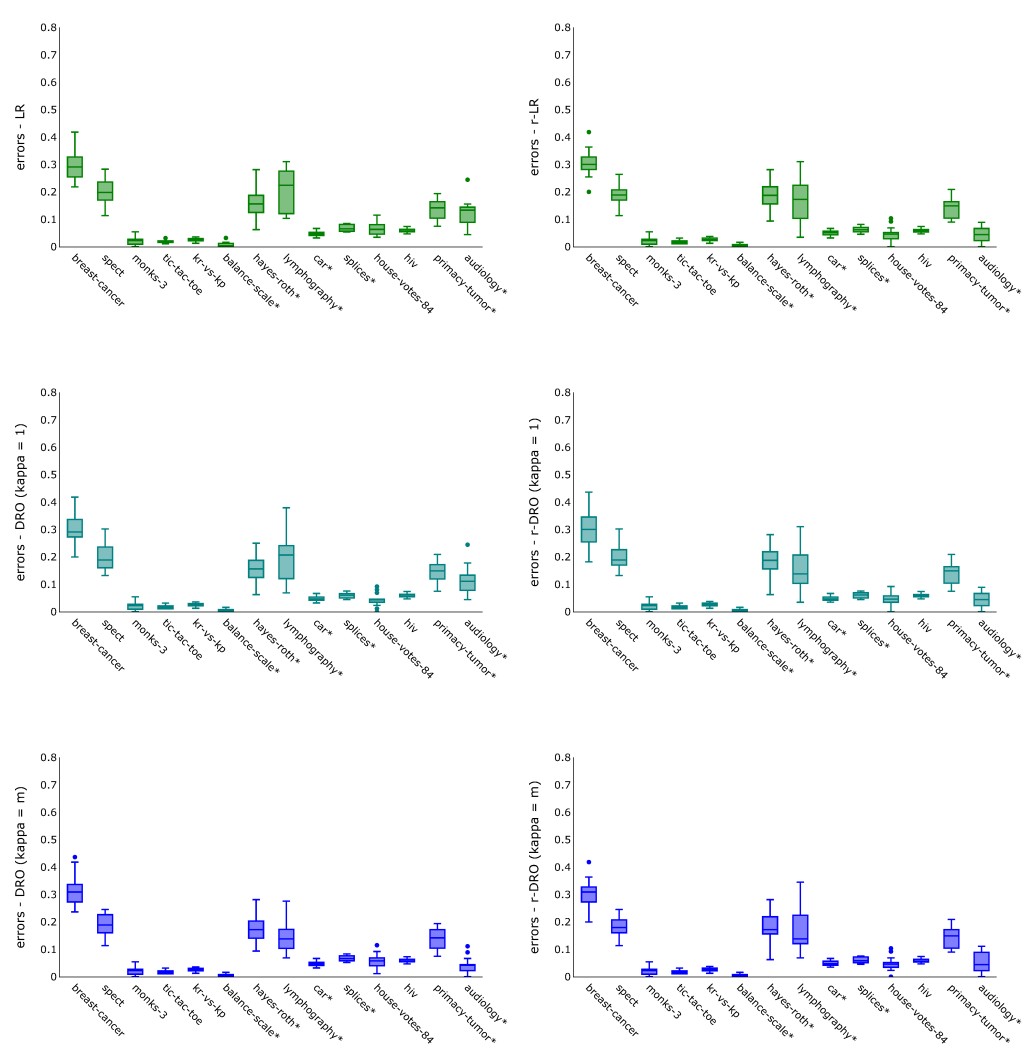

Figure 3: Error bars for the unregularized (left column) and regularized (right column) methods on the considered UCI datasets with categorical features.

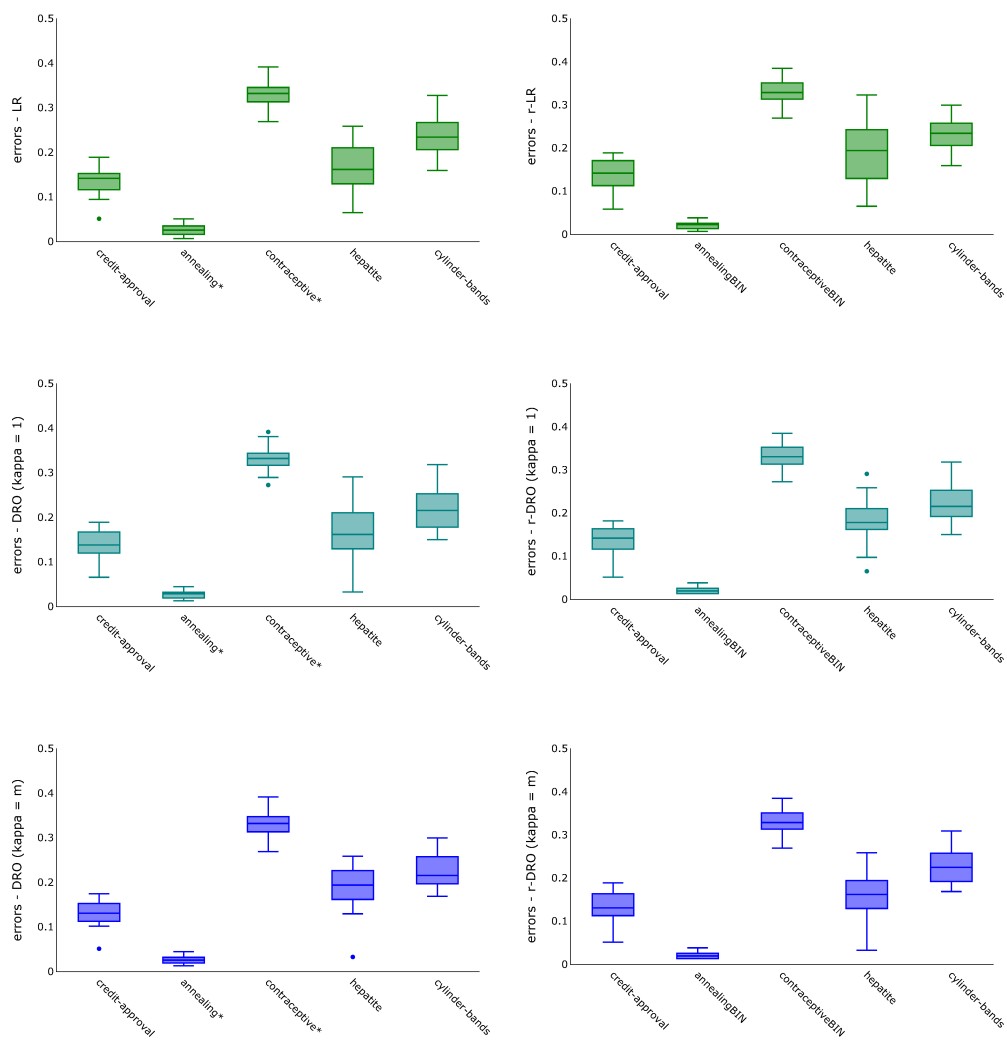

Figure 4: Error bars for the unregularized (left column) and regularized (right column) methods on the considered UCI datasets with mixed features.

# Appendix E: Proofs

Our proof of Theorem 1 relies on the following result from the literature, which we state first.

**Lemma 2.** *Consider the convex function $h_{\boldsymbol{\gamma}}(\boldsymbol{w}) := \log\left(1 + \exp(-\boldsymbol{\gamma}^\top \boldsymbol{w} + v)\right)$ for $\boldsymbol{\gamma}, \boldsymbol{w} \in \mathbb{R}^n$ and $v \in \mathbb{R}$. Then*

$$\sup_{\boldsymbol{w} \in \mathbb{R}^n} h_{\boldsymbol{\gamma}}(\boldsymbol{w}) - \lambda ||\widehat{\boldsymbol{w}} - \boldsymbol{w}|| = \begin{cases} h_{\boldsymbol{\gamma}}(\widehat{\boldsymbol{w}}) & \text{if } ||\boldsymbol{\gamma}||_* \leq \lambda, \\ +\infty & \text{otherwise} \end{cases}$$

*for every $\lambda > 0$, where $|| \cdot ||_*$ is the dual norm of $|| \cdot ||$, that is, $||\boldsymbol{\gamma}||_* := \sup_{||\boldsymbol{w}|| \leq 1} \boldsymbol{\gamma}^\top \boldsymbol{w}$.*

*Proof.* The statement immediately follows from Lemma 47 of [32]. $\qquad\square$

Our proof of Theorem 1 follows [31] with two main changes: *(i)* we handle the categorical features in $\boldsymbol{z}$, and *(ii)* we reformulate the optimization problem as an exponential conic program. Despite the similarities to the existing result, we include the entire proof to keep the paper self-contained.

**Proof of Theorem 1:** Recall that $\boldsymbol{\xi} = (\boldsymbol{x}, \boldsymbol{z}, y) \in \Xi = \mathbb{R}^n \times \mathbb{C} \times \{-1, +1\}$. To simplify our notation, we will use $\boldsymbol{\xi}$ and $(\boldsymbol{x}, \boldsymbol{z}, y)$ interchangeably. The inner problem in (1) can be written as

$$\begin{bmatrix} \underset{\mathbb{Q}}{\text{maximize}} & \mathbb{E}_{\mathbb{Q}}\left[l_{\boldsymbol{\beta}}(\boldsymbol{x}, \boldsymbol{z}, y)\right] \\ \text{subject to} & \mathbb{Q} \in \mathfrak{B}_\epsilon(\widehat{\mathbb{P}}_N) \end{bmatrix} = \begin{bmatrix} \underset{\mathbb{Q}}{\text{maximize}} & \int_{\boldsymbol{\xi} \in \Xi} l_{\boldsymbol{\beta}}(\boldsymbol{\xi})\,\mathbb{Q}(\mathrm{d}\boldsymbol{\xi}) \\ \text{subject to} & \mathbb{Q} \in \mathfrak{B}_\epsilon(\widehat{\mathbb{P}}_N) \end{bmatrix},$$

and replacing $\mathfrak{B}_\epsilon(\widehat{\mathbb{P}}_N)$ with its definition (*cf.* Definition 1) yields

$$\begin{aligned} \underset{\mathbb{Q},\Pi}{\text{maximize}} \quad & \int_{\boldsymbol{\xi} \in \Xi} l_{\boldsymbol{\beta}}(\boldsymbol{\xi})\,\mathbb{Q}(\mathrm{d}\boldsymbol{\xi}) \\ \text{subject to} \quad & \int_{(\boldsymbol{\xi}, \boldsymbol{\xi}') \in \Xi^2} d(\boldsymbol{\xi}, \boldsymbol{\xi}')\,\Pi(\mathrm{d}\boldsymbol{\xi}, \mathrm{d}\boldsymbol{\xi}') \leq \epsilon \\ & \int_{\boldsymbol{\xi} \in \Xi} \Pi(\mathrm{d}\boldsymbol{\xi}, \mathrm{d}\boldsymbol{\xi}') = \widehat{\mathbb{P}}_N(\mathrm{d}\boldsymbol{\xi}') \qquad \forall \boldsymbol{\xi}' \in \Xi \\ & \int_{\boldsymbol{\xi}' \in \Xi} \Pi(\mathrm{d}\boldsymbol{\xi}, \mathrm{d}\boldsymbol{\xi}') = \mathbb{Q}(\mathrm{d}\boldsymbol{\xi}) \qquad \forall \boldsymbol{\xi} \in \Xi \\ & \mathbb{Q} \in \mathcal{P}_0(\Xi), \;\; \Pi \in \mathcal{P}_0(\Xi^2). \end{aligned}$$

We can substitute $\mathbb{Q}$ using the second equality constraint to obtain

$$\begin{aligned} \underset{\Pi}{\text{maximize}} \quad & \int_{\boldsymbol{\xi} \in \Xi} l_{\boldsymbol{\beta}}(\boldsymbol{\xi}) \int_{\boldsymbol{\xi}' \in \Xi} \Pi(\mathrm{d}\boldsymbol{\xi}, \mathrm{d}\boldsymbol{\xi}') \\ \text{subject to} \quad & \int_{(\boldsymbol{\xi}, \boldsymbol{\xi}') \in \Xi^2} d(\boldsymbol{\xi}, \boldsymbol{\xi}')\,\Pi(\mathrm{d}\boldsymbol{\xi}, \mathrm{d}\boldsymbol{\xi}') \leq \epsilon \\ & \int_{\boldsymbol{\xi} \in \Xi} \Pi(\mathrm{d}\boldsymbol{\xi}, \mathrm{d}\boldsymbol{\xi}') = \widehat{\mathbb{P}}_N(\mathrm{d}\boldsymbol{\xi}') \qquad \forall \boldsymbol{\xi}' \in \Xi \\ & \Pi \in \mathcal{P}_0(\Xi^2). \end{aligned}$$

Denoting by $\mathbb{Q}^i(\mathrm{d}\boldsymbol{\xi}) := \Pi(\mathrm{d}\boldsymbol{\xi}|\boldsymbol{\xi}^i)$ the conditional distribution of $\Pi$ upon the realization of $\boldsymbol{\xi}' = \boldsymbol{\xi}^i$ and exploiting the fact that $\widehat{\mathbb{P}}_N$ is a discrete distribution supported on the $N$ atoms $\{\boldsymbol{\xi}^i\}_{i \in [N]}$, we can use the marginalized representation $\Pi(\mathrm{d}\boldsymbol{\xi}, \mathrm{d}\boldsymbol{\xi}') = \frac{1}{N} \sum_{i=1}^N \delta_{\boldsymbol{\xi}^i}(\mathrm{d}\boldsymbol{\xi}')\mathbb{Q}^i(\mathrm{d}\boldsymbol{\xi})$ to obtain the equivalent reformulation

$$\begin{aligned} \underset{\mathbb{Q}^i}{\text{maximize}} \quad & \frac{1}{N} \sum_{i=1}^N \int_{\boldsymbol{\xi} \in \Xi} l_{\boldsymbol{\beta}}(\boldsymbol{\xi})\,\mathbb{Q}^i(\mathrm{d}\boldsymbol{\xi}) \\ \text{subject to} \quad & \frac{1}{N} \sum_{i=1}^N \int_{\boldsymbol{\xi} \in \Xi} d(\boldsymbol{\xi}, \boldsymbol{\xi}^i)\,\mathbb{Q}^i(\mathrm{d}\boldsymbol{\xi}) \leq \epsilon \\ & \mathbb{Q}^i \in \mathcal{P}_0(\Xi), \; i \in [N]. \end{aligned}$$

We can now decompose the $N$ variables (distributions) $\mathbb{Q}^i$ into $2N \cdot \prod_j k_j$ measures $\mathbb{Q}^i_{\boldsymbol{z},y}$:

$$\underset{\mathbb{Q}^i_{\boldsymbol{z},y}}{\text{maximize}} \quad \frac{1}{N} \sum_{i=1}^{N} \sum_{(\boldsymbol{z},y) \in \mathbb{C} \times \{-1,+1\}} \int_{\boldsymbol{x} \in \mathbb{R}^n} l_{\boldsymbol{\beta}}(\boldsymbol{\xi}) \, \mathbb{Q}^i_{\boldsymbol{z},y}(\mathrm{d}\boldsymbol{x})$$

$$\text{subject to} \quad \frac{1}{N} \sum_{i=1}^{N} \sum_{(\boldsymbol{z},y) \in \mathbb{C} \times \{-1,+1\}} \int_{\boldsymbol{x} \in \mathbb{R}^n} d(\boldsymbol{\xi}, \boldsymbol{\xi}^i) \, \mathbb{Q}^i_{\boldsymbol{z},y}(\mathrm{d}\boldsymbol{x}) \leq \epsilon$$

$$\sum_{(\boldsymbol{z},y) \in \mathbb{C} \times \{-1,+1\}} \int_{\boldsymbol{x} \in \mathbb{R}^n} \mathbb{Q}^i_{\boldsymbol{z},y}(\mathrm{d}\boldsymbol{x}) = 1 \qquad \forall i \in [N]$$

$$\mathbb{Q}^i_{\boldsymbol{z},y} \in \mathcal{M}_+(\mathbb{R}^n), \, i \in [N] \text{ and } (\boldsymbol{z},y) \in \mathbb{C} \times \{-1,+1\},$$

where $\mathcal{M}_+(\mathbb{R}^n)$ denotes the space of non-negative measures supported on $\mathbb{R}^n$. This infinite-dimensional linear program admits the dual

$$\underset{\lambda,\boldsymbol{u}}{\text{minimize}} \quad \lambda \epsilon + \sum_{i=1}^{N} u_i$$

$$\text{subject to} \quad \frac{1}{N} \cdot \sup_{\boldsymbol{x} \in \mathbb{R}^n} \left\{ l_{\boldsymbol{\beta}}(\boldsymbol{x},\boldsymbol{z},y) - \lambda d((\boldsymbol{x},\boldsymbol{z},y), \boldsymbol{\xi}^i) \right\} \leq u_i \qquad \forall i \in [N]$$

$$\forall (\boldsymbol{z},y) \in \mathbb{C} \times \{-1,+1\}$$

$$\lambda \geq 0, \ \boldsymbol{u} \in \mathbb{R}^N.$$

Strong duality holds for any $\epsilon > 0$ due to Proposition 3.4 of [33]. We rewrite the dual problem further by substituting $s_i := N u_i$, using Definition 2 to write out $d$ explicitly, as well as breaking down the dual constraints into separate cases for $y = -1$ and $y = +1$:

$$\underset{\lambda,\boldsymbol{s}}{\text{minimize}} \quad \lambda \epsilon + \frac{1}{N} \sum_{i=1}^{N} s_i$$

$$\text{subject to} \quad \sup_{\boldsymbol{x} \in \mathbb{R}^n} \left\{ l_{\boldsymbol{\beta}}(\boldsymbol{x},\boldsymbol{z},+1) - \lambda \|\boldsymbol{x} - \boldsymbol{x}^i\| \right\} - \lambda \kappa \cdot \mathbb{1}[y^i \neq 1] - \lambda d_{\mathrm{C}}(\boldsymbol{z}, \boldsymbol{z}^i) \leq s_i \qquad \forall i \in [N], \ \boldsymbol{z} \in \mathbb{C}$$

$$\sup_{\boldsymbol{x} \in \mathbb{R}^n} \left\{ l_{\boldsymbol{\beta}}(\boldsymbol{x},\boldsymbol{z},-1) - \lambda \|\boldsymbol{x} - \boldsymbol{x}^i\| \right\} - \lambda \kappa \cdot \mathbb{1}[y^i \neq -1] - \lambda d_{\mathrm{C}}(\boldsymbol{z}, \boldsymbol{z}^i) \leq s_i \qquad \forall i \in [N], \ \boldsymbol{z} \in \mathbb{C}$$

$$\lambda \geq 0, \ \boldsymbol{s} \in \mathbb{R}^N$$

Applying Lemma 2 to the suprema of the above problem results in the reformulation

$$\underset{\lambda,\boldsymbol{s}}{\text{minimize}} \quad \lambda \epsilon + \frac{1}{N} \sum_{i=1}^{N} s_i$$

$$\text{subject to} \quad l_{\boldsymbol{\beta}}(\boldsymbol{x}^i,\boldsymbol{z},+1) - \lambda \kappa \cdot \mathbb{1}[y^i \neq 1] - \lambda d_{\mathrm{C}}(\boldsymbol{z}, \boldsymbol{z}^i) \leq s_i \qquad \forall i \in [N], \ \forall \boldsymbol{z} \in \mathbb{C}$$

$$l_{\boldsymbol{\beta}}(\boldsymbol{x}^i,\boldsymbol{z},-1) - \lambda \kappa \cdot \mathbb{1}[y^i \neq -1] - \lambda d_{\mathrm{C}}(\boldsymbol{z}, \boldsymbol{z}^i) \leq s_i \qquad \forall i \in [N], \ \forall \boldsymbol{z} \in \mathbb{C}$$

$$\|\boldsymbol{\beta}_{\mathrm{N}}\|_* \leq \lambda$$

$$\lambda \geq 0, \ \boldsymbol{s} \in \mathbb{R}^N.$$

For $y^i = +1$, the identities

$$l_{\boldsymbol{\beta}}(\boldsymbol{x}^i,\boldsymbol{z},+1) - \lambda \kappa \cdot \mathbb{1}[y^i \neq 1] = l_{\boldsymbol{\beta}}(\boldsymbol{x}^i,\boldsymbol{z},y^i)$$

$$l_{\boldsymbol{\beta}}(\boldsymbol{x}^i,\boldsymbol{z},-1) - \lambda \kappa \cdot \mathbb{1}[y^i \neq -1] = l_{\boldsymbol{\beta}}(\boldsymbol{x}^i,\boldsymbol{z},-y^i) - \lambda \kappa$$

hold; similar identities hold for $y^i = -1$. The above optimization problem thus simplifies to

$$\underset{\lambda,\boldsymbol{s}}{\text{minimize}} \quad \lambda \epsilon + \frac{1}{N} \sum_{i=1}^{N} s_i$$

$$\text{subject to} \quad l_{\boldsymbol{\beta}}(\boldsymbol{x}^i,\boldsymbol{z},y^i) - \lambda d_{\mathrm{C}}(\boldsymbol{z}, \boldsymbol{z}^i) \leq s_i \qquad \forall i \in [N], \ \forall \boldsymbol{z} \in \mathbb{C} \qquad (9)$$

$$l_{\boldsymbol{\beta}}(\boldsymbol{x}^i,\boldsymbol{z},-y^i) - \lambda \kappa - \lambda d_{\mathrm{C}}(\boldsymbol{z}, \boldsymbol{z}^i) \leq s_i \qquad \forall i \in [N], \ \forall \boldsymbol{z} \in \mathbb{C}$$

$$\|\boldsymbol{\beta}_{\mathrm{N}}\|_* \leq \lambda$$

$$\lambda \geq 0, \ \boldsymbol{s} \in \mathbb{R}^N.$$

Plugging this problem into the overall optimization problem over $\boldsymbol{\beta} \in \mathbb{R}^{1+n+k}$ therefore yields

$$
\begin{aligned}
\underset{\boldsymbol{\beta}, \lambda, \boldsymbol{s}}{\text{minimize}} \quad & \lambda\epsilon + \frac{1}{N}\sum_{i=1}^{N} s_i \\
\text{subject to} \quad & l_{\boldsymbol{\beta}}(\boldsymbol{x}^i, \boldsymbol{z}, y^i) - \lambda d_{\mathrm{C}}(\boldsymbol{z}, \boldsymbol{z}^i) \le s_i && \forall i \in [N], \ \forall \boldsymbol{z} \in \mathbb{C} \quad\quad (10)\\
& l_{\boldsymbol{\beta}}(\boldsymbol{x}^i, \boldsymbol{z}, -y^i) - \lambda\kappa - \lambda d_{\mathrm{C}}(\boldsymbol{z}, \boldsymbol{z}^i) \le s_i && \forall i \in [N], \ \forall \boldsymbol{z} \in \mathbb{C} \\
& \|\boldsymbol{\beta}_{\mathrm{N}}\|_* \le \lambda \\
& \boldsymbol{\beta} = (\beta_0, \boldsymbol{\beta}_{\mathrm{N}}, \boldsymbol{\beta}_{\mathrm{C}}) \in \mathbb{R}^{1+n+k}, \ \ \lambda \ge 0, \ \ \boldsymbol{s} \in \mathbb{R}^N.
\end{aligned}
$$

Finally, we discuss how to reformulate the *softplus constraints* (*i.e.*, convex constraints with a log-loss function on the left-hand side and a linear function on the right-hand side) as exponential cone constraints. To this end, recall that the *exponential cone* is defined as

$$
\mathcal{K}_{\exp} := \mathrm{cl}\left(\{(a, b, c) \ : \ a \ge b\exp(c/b), \ a > 0, \ b > 0\}\right) \subset \mathbb{R}^3,
$$

where $\mathrm{cl}(\cdot)$ denotes the closure. Observe further that

$$
\log(1 + \exp(c)) \le a \iff 1 + \exp(c) \le \exp(a) \iff \exp(-a) + \exp(c - a) \le 1.
$$

Using the auxiliary variables $u, v$, we can reformulate this constraint as

$$
\begin{bmatrix} u + v \le 1 \\ \exp(-a) \le u \\ \exp(c - a) \le v \end{bmatrix}
\iff
\begin{bmatrix} u + v \le 1 \\ (u, 1, -a) \in \mathcal{K}_{\exp} \\ (v, 1, c - a) \in \mathcal{K}_{\exp} \end{bmatrix},
$$

where the second system of equations uses the definition of $\mathcal{K}_{\exp}$. Applying this reformulation to both softplus constraints in our minimization problem results in problem (4) and thus concludes the proof of the theorem. $\square$

**Remark 1** (Exponential Cone Reformulation). *The use of exponential conic constraints has a significant impact on the theoretical complexity of problem (4). We refer to [30] for an overview of exponential conic programs and to [26] for modeling techniques, respectively. We also note that the difference in theoretical complexity carries over to a significant difference in practical solvability: In our experiments, we observe that using the exponential cone solver of MOSEK drastically speeds up the solution times of our DR logistic regression problem.*

**Proof of Theorem 2:** In view of the first statement, we recall that the strongly NP-hard integer programming problem is defined as follows [16]:

---
0/1 INTEGER PROGRAMMING.
**Instance.** Given are $\boldsymbol{F} \in \mathbb{Z}^{\mu \times \nu}$, $\boldsymbol{g} \in \mathbb{Z}^\mu$, $\boldsymbol{c} \in \mathbb{Z}^\nu$, $\zeta \in \mathbb{Z}$.
**Question.** Is there a vector $\boldsymbol{\chi} \in \{0, 1\}^\nu$ such that $\boldsymbol{F}\boldsymbol{\chi} \le \boldsymbol{g}$ and $\boldsymbol{c}^\top \boldsymbol{x} \ge \zeta$?

---

Here, $\mathbb{Z}$ denotes the set of integers. We claim that the answer to the integer programming problem is affirmative if and only if the optimal objective value for the DR logistic regression (1) with $n = 0$, $m = \nu$, $k_1 = \ldots = k_m = 2$, $N = 1$ and the loss function

$$
l_{\boldsymbol{\beta}}(\boldsymbol{z}, y) = \begin{cases} \boldsymbol{c}^\top \boldsymbol{z} & \text{if } \boldsymbol{F}\boldsymbol{z} \le \boldsymbol{g}, \\ -\mathrm{M} & \text{otherwise} \end{cases}
$$

and M and $\epsilon$ sufficiently large is greater than or equal to $\zeta$. This then immdiately implies that the DR logistic regression (1) is strongly NP-hard for generic loss functions (which include the above loss function as a special case).

To see this, note that our reformulation from the proof of Theorem 1, under the assumption that $n = 0$ and $N = 1$, shows that problem (1) is equivalent to

$$
\begin{aligned}
\underset{\boldsymbol{\beta}, \lambda, s}{\text{minimize}} \quad & \lambda\epsilon + s \\
\text{subject to} \quad & l_{\boldsymbol{\beta}}(\boldsymbol{z}, y^1) - \lambda d_{\mathrm{C}}(\boldsymbol{z}, \boldsymbol{z}^1) \le s && \forall \boldsymbol{z} \in \mathbb{C} \\
& l_{\boldsymbol{\beta}}(\boldsymbol{z}, -y^1) - \lambda\kappa - \lambda d_{\mathrm{C}}(\boldsymbol{z}, \boldsymbol{z}^1) \le s && \forall \boldsymbol{z} \in \mathbb{C} \\
& \boldsymbol{\beta} = (\beta_0, \boldsymbol{\beta}_{\mathrm{C}}) \in \mathbb{R}^{k+1}, \ \ \lambda \ge 0, \ \ s \in \mathbb{R}.
\end{aligned}
$$

Assuming that $\epsilon$ is sufficiently large, the optimal $\lambda$ vanishes, and the problem simplifies further to

$$\underset{\boldsymbol{\beta},s}{\text{minimize}} \quad s$$

$$\begin{aligned}
\text{subject to} \quad & \boldsymbol{c}^{\top}\boldsymbol{z} \leq s \quad \forall \boldsymbol{z} \in \mathbb{C} : \boldsymbol{F}\boldsymbol{z} \leq \boldsymbol{g} \\
& -\mathrm{M} \leq s \quad \text{if } \exists \boldsymbol{z} \in \mathbb{C} : \boldsymbol{F}\boldsymbol{z} \not\leq \boldsymbol{g} \\
& \boldsymbol{\beta} = (\beta_0, \boldsymbol{\beta}_{\mathrm{C}}) \in \mathbb{R}^{k+1}, \quad s \in \mathbb{R}
\end{aligned}$$

where we have also plugged in the aforementioned loss function. One readily observes that the optimal value of this problem equals $\max\{\boldsymbol{c}^{\top}\boldsymbol{\chi} : \boldsymbol{F}\boldsymbol{\chi} \leq \boldsymbol{g}, \ \boldsymbol{\chi} \in \{0,1\}^{\nu}\}$ whenever $\{\boldsymbol{F}\boldsymbol{\chi} \leq \boldsymbol{g}, \ \boldsymbol{\chi} \in \{0,1\}^{\nu}\}$ is non-empty and $-\mathrm{M}$ otherwise. In particular, the optimal value of the problem is greater than or equal to $\zeta$ if and only if the answer to the integer programming problem is affirmative.

In view of the second statement, we note that the feasible region of problem (1) can be circumscribed by a convex body as per Definition 2.1.16 of [17], and that Algorithm 2 in Section 3 constitutes a weak separation oracle for problem (1). The statement thus follows from Corollary 4.2.7 of [17]. $\qquad\square$

**Proof of Theorem 3:** Consider a class of instances of problem (1) with a Wasserstein radius $\epsilon < 1$, $n = 1$ numerical feature, and $m = 1$ categorical feature with $k_1 = 2$ so that the categorical feature is binary. We have $N = 1$ sample with $z_1^1 = 0$ and $y_1^1 = -1$. This instance class is therefore parameterized by the value of the numerical feature $x_1^1$ in the data sample. We now derive the optimal objective value of problem (1) analytically as a function of $x_1^1$, and show that this functional dependence on $x_1^1$ cannot be represented by the objective function of *any* regularized logistic regression.

For our instance class, the reformulation of our DR logistic regression (1) simplifies to

$$\begin{aligned}
\underset{\lambda, s_1, \boldsymbol{\beta}}{\text{minimize}} \quad & \lambda\epsilon + s_1 \\
\text{subject to} \quad & l_{\boldsymbol{\beta}}(x_1^1, z, -1) - \lambda z \leq s_1 \qquad \forall z \in \mathbb{B} \\
& l_{\boldsymbol{\beta}}(x_1^1, z, 1) - \lambda\kappa - \lambda z \leq s_1 \qquad \forall z \in \mathbb{B} \\
& \lambda \geq |\boldsymbol{\beta}_{\mathrm{N}}|, \quad s_1 \in \mathbb{R}.
\end{aligned} \tag{11}$$

Assuming further that $\kappa$ approaches $\infty$, the constraint $\lambda \geq 0$ (implied by the constraint $\lambda \geq |\boldsymbol{\beta}_{\mathrm{N}}|$) implies that the second constraint is redundant. Removing this second constraint and substituting $s_1$ into the objective function results in

$$\begin{aligned}
\underset{\lambda, \boldsymbol{\beta}}{\text{minimize}} \quad & \lambda\epsilon + \max\left\{l_{\boldsymbol{\beta}}(x_1^1, 0, -1), \ l_{\boldsymbol{\beta}}(x_1^1, 1, -1) - \lambda\right\} \\
\text{subject to} \quad & \lambda \geq |\boldsymbol{\beta}_{\mathrm{N}}|.
\end{aligned} \tag{12}$$

We now break the minimization over $(\lambda, \boldsymbol{\beta})$ in (12) into a minimization over $\lambda$ first followed by the minimization over $\boldsymbol{\beta}$. That leads to re-writing (12) as

$$\underset{\boldsymbol{\beta}}{\text{minimize}} \quad f(\boldsymbol{\beta}, x_1^1) \tag{13}$$

where

$$f(\boldsymbol{\beta}, x_1^1) = \left\{ \begin{aligned}
\underset{\lambda}{\text{minimize}} \quad & \lambda\epsilon + \max\left\{l_{\boldsymbol{\beta}}(x_1^1, 0, -1), \ l_{\boldsymbol{\beta}}(x_1^1, 1, -1) - \lambda\right\} \\
\text{subject to} \quad & \lambda \geq |\boldsymbol{\beta}_{\mathrm{N}}|
\end{aligned} \right. . \tag{14}$$

Since the Wasserstein radius satisfies $\epsilon < 1$, it is possible to achieve a reduction in the objective function choosing values larger than $|\boldsymbol{\beta}_{\mathrm{N}}|$ for $\lambda$. Focusing on the minimization over $\lambda$ in (14), we recognize that the optimal solution is $\lambda^{\star} = g(\boldsymbol{\beta}, x_1^1) := l_{\boldsymbol{\beta}}(x_1^1, 1, -1) - l_{\boldsymbol{\beta}}(x_1^1, 0, -1)$ if $g(\boldsymbol{\beta}, x_1^1) \geq |\boldsymbol{\beta}_{\mathrm{N}}|$, and $\lambda^{\star} = |\boldsymbol{\beta}_{\mathrm{N}}|$ otherwise. Substituting this optimal value for $\lambda$ into (14), we obtain

$$\begin{aligned}
f(\boldsymbol{\beta}, x_1^1) &= \begin{cases} \epsilon \cdot g(\boldsymbol{\beta}, x_1^1) + l_{\boldsymbol{\beta}}(x_1^1, 0, -1) & \text{if } |\boldsymbol{\beta}_{\mathrm{N}}| \leq g(\boldsymbol{\beta}, x_1^1) \\ \epsilon \cdot |\boldsymbol{\beta}_{\mathrm{N}}| + l_{\boldsymbol{\beta}}(x_1^1, 0, -1) & \text{otherwise} \end{cases} \\
&= l_{\boldsymbol{\beta}}(x_1^1, 0, -1) + h(\boldsymbol{\beta}, x_1^1),
\end{aligned}$$

where

$$h(\boldsymbol{\beta}, x_1^1) := \epsilon \cdot \begin{cases} g(\boldsymbol{\beta}, x_1^1) & \text{if } |\boldsymbol{\beta}_{\mathrm{N}}| \leq g(\boldsymbol{\beta}, x_1^1) \\ |\boldsymbol{\beta}_{\mathrm{N}}| & \text{otherwise.} \end{cases} = \epsilon \cdot \max\left\{g(\boldsymbol{\beta}, x_1^1), \ |\boldsymbol{\beta}_{\mathrm{N}}|\right\}.$$

The DR logistic regression problem (13) now becomes

$$\underset{\boldsymbol{\beta}}{\text{minimize}} \quad l_{\boldsymbol{\beta}}(x_1^1, 0, -1) + h(\boldsymbol{\beta}, x_1^1).$$

Using the definition of the empirical distribution, the regularized logistic regression for the instance problem has the form

$$\underset{\boldsymbol{\beta}}{\text{minimize}} \quad l_{\boldsymbol{\beta}}(x_1^1, 0, -1) + \mathfrak{R}(\boldsymbol{\beta}). \tag{15}$$

The proof concludes by noting that $\mathfrak{R}(\boldsymbol{\beta})$ remains constant across all the instances parameterized by $x_1^1$ as a regularizer is data-independent. Hence, the objective function of the regularized logistic regression cannot capture the dependency on $x_1^1$ that we observe in $h(\boldsymbol{\beta}, x_1^1)$ (it is easy to confirm that $h(\boldsymbol{\beta}, x_1^1)$ is not constant in $x_1^1$). We further notice that the objective function of DR logistic regression is a non-smooth function of $x_1^1$ while the objective function of regularized logistic regression is a smooth function of $x_1^1$ by construction. $\qquad \square$

**Remark 2.** *It is informative to see how the counter-example in the proof of Theorem 3 breaks down if we assume there are no categorical features. In this case we use the ground metric $d(\boldsymbol{\xi}, \boldsymbol{\xi}') := \|\boldsymbol{x} - \boldsymbol{x}'\| + \kappa \cdot \mathbb{1}[y \neq y']$. The constraints in the problem formulation (10) will no longer be indexed over $\boldsymbol{z}$ and all terms involving $\boldsymbol{z}$ and $d_C(\boldsymbol{z}, \boldsymbol{z}^i)$ will disappear. The reformulation with only numerical features then coincides with that of Corollary 17 in [32]. Our instance class problem from the proof of Theorem 3 above will then be*

$$\begin{aligned}
\underset{\lambda, s_1, \boldsymbol{\beta}}{\text{minimize}} \quad & \lambda\epsilon + s_1 \\
\text{subject to} \quad & l_{\boldsymbol{\beta}}(x_1^1, -1) \leq s_1 \\
& l_{\boldsymbol{\beta}}(x_1^1, 1) - \lambda\kappa \leq s_1 \\
& \lambda \geq |\boldsymbol{\beta}_{\mathrm{N}}|, \quad s_1 \in \mathbb{R}.
\end{aligned}$$

*Taking $\kappa$ to infinity, we see the second constraint becomes redundant. As $\lambda$ only appears in one term in the objective function, the optimal solution is $\lambda^\star = |\boldsymbol{\beta}_N|$. Substituting for $s_1$ in the resulting objective becomes*

$$\underset{\boldsymbol{\beta}}{\text{minimize}} \quad l_{\boldsymbol{\beta}}(x_1^1, -1) + \epsilon|\boldsymbol{\beta}_{\mathrm{N}}|$$

*which is in the form of regularized logistic regression (15). This of course is consistent with [31] who showed that DR logistic regression without categorical features could be formulated as a regularized logistic regression problem.*

**Proof of Theorem 4:** It follows from the proof of Theorem 1 that the optimization problems solved inside the while-loop of Algorithm 1 are equivalent to the relaxations

$$\begin{aligned}
\underset{\boldsymbol{\beta}, \lambda, \boldsymbol{s}}{\text{minimize}} \quad & \lambda\epsilon + \frac{1}{N}\sum_{i=1}^{N} s_i \\
\text{subject to} \quad & l_{\boldsymbol{\beta}}(\boldsymbol{x}^i, \boldsymbol{z}, y^i) - \lambda d_{\mathrm{C}}(\boldsymbol{z}, \boldsymbol{z}^i) \leq s_i & \forall(i, \boldsymbol{z}) \in \mathcal{W}^+ \\
& l_{\boldsymbol{\beta}}(\boldsymbol{x}^i, \boldsymbol{z}, -y^i) - \lambda\kappa - \lambda d_{\mathrm{C}}(\boldsymbol{z}, \boldsymbol{z}^i) \leq s_i & \forall(i, \boldsymbol{z}) \in \mathcal{W}^- \\
& \|\boldsymbol{\beta}_{\mathrm{N}}\|_* \leq \lambda \\
& \boldsymbol{\beta} = (\beta_0, \boldsymbol{\beta}_{\mathrm{N}}, \boldsymbol{\beta}_{\mathrm{C}}) \in \mathbb{R}^{1+n+k}, \quad \lambda \geq 0, \quad \boldsymbol{s} \in \mathbb{R}^N
\end{aligned}$$

of problem (10) that is itself equivalent to our DR logistic regression problem (4); the only difference between our relaxations above and problem (10) is that the index sets $(i, \boldsymbol{z}) \in [N] \times \mathbb{C}$ in the first two constraint sets of (10) are replaced with the subsets $\mathcal{W}^+$ and $\mathcal{W}^-$ above. This shows that each value $\mathrm{LB}_t$ indeed constitutes a lower bound on the optimal value of problem (4), and that the sequence $\{\mathrm{LB}_t\}_t$ of lower bounds is monotonically non-decreasing.

To see that the sequence $\{\mathrm{UB}_t\}_t$ bounds the optimal value of our DR logistic regression problem (4) from above, note that a maximum constraint violation of $\vartheta^+$ in the first constraint set of (4) implies that

$$\left[\begin{array}{l}
u_{i,\boldsymbol{z}}^+ + v_{i,\boldsymbol{z}}^+ \leq 1 + \vartheta^+ \\
(u_{i,\boldsymbol{z}}^+, 1, -s_i - \lambda d_{\mathrm{C}}(\boldsymbol{z}, \boldsymbol{z}^i)) \in \mathcal{K}_{\exp} \\
(v_{i,\boldsymbol{z}}^+, 1, -y^i \boldsymbol{\beta}_{\mathrm{N}}^\top \boldsymbol{x}^i - y^i \boldsymbol{\beta}_{\mathrm{C}}^\top \boldsymbol{z} - y^i \beta_0 - s_i - \lambda d_{\mathrm{C}}(\boldsymbol{z}, \boldsymbol{z}^i)) \in \mathcal{K}_{\exp},
\end{array}\right] \quad \forall(i, \boldsymbol{z}) \in [N] \times \mathbb{C},$$

and, by the definition of the exponential cone, this means that for all $(i, \boldsymbol{z}) \in [N] \times \mathbb{C}$, we have

$$\exp\left[-s_i - \lambda d_{\mathrm{C}}(\boldsymbol{z}, \boldsymbol{z}^i)\right] + \exp\left[-y^i \boldsymbol{\beta}_{\mathrm{N}}^\top \boldsymbol{x}^i - y^i \boldsymbol{\beta}_{\mathrm{C}}^\top \boldsymbol{z} - y^i \beta_0 - s_i - \lambda d_{\mathrm{C}}(\boldsymbol{z}, \boldsymbol{z}^i)\right] \leq 1 + \vartheta^+.$$

Dividing both sides by $1 + \vartheta^+ = \exp[\log(1 + \vartheta^+)]$, the inequality becomes

$$\exp\left[-s_i' - \lambda d_{\mathrm{C}}(\boldsymbol{z}, \boldsymbol{z}^i)\right] + \exp\left[-y^i \boldsymbol{\beta}_{\mathrm{N}}^\top \boldsymbol{x}^i - y^i \boldsymbol{\beta}_{\mathrm{C}}^\top \boldsymbol{z} - y^i \beta_0 - s_i' - \lambda d_{\mathrm{C}}(\boldsymbol{z}, \boldsymbol{z}^i)\right] \leq 1,$$

where $s_i' := s_i + \log(1 + \vartheta^+)$. The (potentially) infeasible solution to the relaxed problem of Algorithm 1 thus allows us to construct a solution to problem (4) that satisfies all members of the first constraint set by replacing $s_i$ with $s_i' := s_i + \log(1 + \vartheta^+)$, $i \in [N]$. Compared to the solution of the relaxed problem, the objective value of the newly created solution increased by $\log(1 + \vartheta^+)$. A similar argument can be made for the members of the second constraint set, which shows that a solution to problem (4) satisfying both constraint sets can be constructed by increasing the objective value by no more than $\log(1 + \max\{\vartheta^+, \vartheta^-\})$. This shows the validity of the upper bound $\theta^\star + \log(1 + \max\{\vartheta^+, \vartheta^-\})$. The fact that the sequence $\{\mathrm{UB}_t\}_t$ of upper bounds is monotonically non-increasing, on the other hand, holds by construction since we take the minimum of the derived upper bound $(\theta^\star + \log(1 + \max\{\vartheta^+, \vartheta^-\})$ and the upper bound $\mathrm{UB}_{t-1}$ of the previous iteration.

To see that Algorithm 1 terminates in finite time, note that every iteration $t$ either identifies a violated constraint $(i, \boldsymbol{z}) \in [([N] \times \mathbb{C}) \setminus \mathcal{W}^+] \cup [([N] \times \mathbb{C}) \setminus \mathcal{W}^-]$ that is subsequently added to $\mathcal{W}^+$ and/or $\mathcal{W}^-$ (and will thus never again be identified as violated for the same constraint set), or the update $\mathrm{LB}_t = \mathrm{UB}_t = \theta^\star$ is conducted, in which case the algorithm terminates in the next iteration. Finite termination thus holds since the set $[N] \times \mathbb{C}$ of potentially violated constraints is finite. □

**Additional Intuition Behind Algorithm 2**

In the combinatorial optimization problem (5), the decision variables $\boldsymbol{z} \in \mathbb{C}$ only appear in (monotone transformations of) the terms $-y^i \boldsymbol{\beta}_{\mathrm{C}}^\top \boldsymbol{z}$ and $-\lambda d_{\mathrm{C}}(\boldsymbol{z}, \boldsymbol{z}^i)$. Moreover, the expression $[d_{\mathrm{C}}(\boldsymbol{z}, \boldsymbol{z}^i)]^p$ can only attain one of the values $\delta \in \{0, \dots, m\}$ (*cf.* Definition 2). Conditioning on each possible value of $\delta$, which records the number of categorical features along which $\boldsymbol{z}$ and $\boldsymbol{z}^i$ disagree, we can therefore maximize the constraint violation by maximizing $-y^i \boldsymbol{\beta}_{\mathrm{C}}^\top \boldsymbol{z}$ along all $\boldsymbol{z} \in \mathbb{C}$ that differ from $\boldsymbol{z}^i$ in exactly $\delta$ categorical features. Finally, since $-y^i \boldsymbol{\beta}_{\mathrm{C}}^\top \boldsymbol{z}$ is linearly separable, that is, $-y^i \boldsymbol{\beta}_{\mathrm{C}}^\top \boldsymbol{z} = -y^i \boldsymbol{\beta}_{\mathrm{C},1}^\top \boldsymbol{z}_1 - \dots - y^i \boldsymbol{\beta}_{\mathrm{C},m}^\top \boldsymbol{z}_m$, we can choose the categorical features along which $\boldsymbol{z}$ and $\boldsymbol{z}^i$ disagree iteratively by considering each expression $-y^i \boldsymbol{\beta}_{\mathrm{C}}^\top \boldsymbol{z}$ separately and choosing the disagreeing feature values greedily based on their individual contribution to the overall constraint violation.

**Proof of Theorem 5** Recall that Algorithm 2 aims to solve the optimization problem

$$\underset{\boldsymbol{z}}{\text{maximize}} \quad \underset{u^+, v^+}{\min} \left\{ u^+ + v^+ \; : \; \begin{bmatrix} (u^+, 1, -s_i - \lambda d_{\mathrm{C}}(\boldsymbol{z}, \boldsymbol{z}^i)) \in \mathcal{K}_{\exp} \\ (v^+, 1, -y^i \boldsymbol{\beta}_{\mathrm{N}}^\top \boldsymbol{x}^i - y^i \boldsymbol{\beta}_{\mathrm{C}}^\top \boldsymbol{z} - y^i \beta_0 - s_i - \lambda d_{\mathrm{C}}(\boldsymbol{z}, \boldsymbol{z}^i)) \in \mathcal{K}_{\exp} \end{bmatrix} \right\}$$

subject to $\quad \boldsymbol{z} \in \mathbb{C}.$

It immediately follows from the definition of $\mathcal{K}_{\exp}$ that the above problem can be represented as

$$\underset{\boldsymbol{z}}{\text{maximize}} \quad \exp\left[-s_i - \lambda d_{\mathrm{C}}(\boldsymbol{z}, \boldsymbol{z}^i)\right] + \exp\left[-y^i \boldsymbol{\beta}_{\mathrm{N}}^\top \boldsymbol{x}^i - y^i \boldsymbol{\beta}_{\mathrm{C}}^\top \boldsymbol{z} - y^i \beta_0 - s_i - \lambda d_{\mathrm{C}}(\boldsymbol{z}, \boldsymbol{z}^i)\right]$$

subject to $\quad \boldsymbol{z} \in \mathbb{C}.$

In this problem, the optimization variable $\boldsymbol{z}$ appears inside the exponential terms as $-y^i \boldsymbol{\beta}_{\mathrm{C}}^\top \boldsymbol{z}$ and as $-\lambda d_{\mathrm{C}}(\boldsymbol{z}, \boldsymbol{z}^i)$. Since increasing the value of one term may decrease the value of the other term, the above problem does not admit a trivial solution. We next discuss how the above maximization problem can be decomposed into $m + 1$ sub-problems, each of which can be solved efficiently.

Although $\boldsymbol{z}$ may take exponentially many values, notice that the expression $d_{\mathrm{C}}(\boldsymbol{z}, \boldsymbol{z}^i) = (\sum_{j \in [m]} \mathbb{1}[\boldsymbol{z}_j \neq \boldsymbol{z}_j^i])^{1/p}$ counts the number of disagreements between the features of $\boldsymbol{z}$ and $\boldsymbol{z}^i$, which takes a value from the set $\{0, 1, \dots, m\}$. Hence, Algorithm 2 decomposes the above combinatorial optimization problem into $m + 1$ sub-problems in which the number $\delta \in \{0, 1, \dots, m\}$ of disagreements between $\boldsymbol{z}$ and $\boldsymbol{z}^i$ is fixed and we only need to maximize $-y^i \boldsymbol{\beta}_{\mathrm{C}}^\top \boldsymbol{z}$ for a given number $\delta$ of disagreements. If we can solve each of these sub-problem efficiently, then we can compare the constraint violations of the optimal $\boldsymbol{z}$ to each sub-problem and pick the largest one as

the most-violated constraint. We next investigate how each of the $m + 1$ sub-problems can be solved efficiently.

The sub-problem corresponding to a fixed $\delta \in \{0, 1, \ldots, m\}$ can be formulated as

$$\begin{aligned}
\underset{\boldsymbol{z}}{\text{maximize}} \quad & -y^i {\boldsymbol{\beta}_{\text{C}}}^\top \boldsymbol{z} \\
\text{subject to} \quad & d_{\text{C}}(\boldsymbol{z}, \boldsymbol{z}^i) = \delta \\
& \boldsymbol{z} \in \mathbb{C}.
\end{aligned}$$

Denote by $\mathcal{J} \subseteq [m] : |\mathcal{J}| = \delta$ the set of features where $\boldsymbol{z}$ disagrees with $\boldsymbol{z}^i$ at optimality. Denote further by $\boldsymbol{z}_j^\star$ the maximizer of $-y^i \cdot {\boldsymbol{\beta}_{\text{C},j}}^\top \boldsymbol{z}_j$ over $\boldsymbol{z}_j \in \mathbb{C}(k_j) \backslash \{\boldsymbol{z}_j^i\}$ (cf. the first step of Algorithm 2). The above sub-problem can then be written as

$$\begin{aligned}
\underset{\mathcal{J}}{\text{maximize}} \quad & \left( \sum_{j=1}^m -y^i \cdot {\boldsymbol{\beta}_{\text{C},j}}^\top \boldsymbol{z}_j^i \right) + \left( \sum_{j \in \mathcal{J}} (-y^i \cdot {\boldsymbol{\beta}_{\text{C},j}}^\top \boldsymbol{z}_j^\star) - (-y^i \cdot {\boldsymbol{\beta}_{\text{C},j}}^\top \boldsymbol{z}_j^i) \right) \\
\text{subject to} \quad & \mathcal{J} \subseteq [m], \ |\mathcal{J}| = \delta.
\end{aligned}$$

The first summation above is constant, and the solution of this problem can therefore be uniquely identified from the solution of the following problem:

$$\begin{aligned}
\underset{\mathcal{J}}{\text{maximize}} \quad & \sum_{j \in \mathcal{J}} (-y^i \cdot {\boldsymbol{\beta}_{\text{C},j}}^\top (\boldsymbol{z}_j^\star - \boldsymbol{z}_j^i)) \\
\text{subject to} \quad & \mathcal{J} \subseteq [m], \ |\mathcal{J}| = \delta.
\end{aligned}$$

This problem can be solved greedily by selecting the $\delta$ largest values of $(-y^i \cdot {\boldsymbol{\beta}_{\text{C},j}}^\top (\boldsymbol{z}_j^\star - \boldsymbol{z}_j^i))$ across all $j \in [m]$, and those $j$ indexes will be included in $\mathcal{J}$ (cf. the ordering $\pi : [m] \mapsto [m]$ and the selection according to $\pi(j) \leq \delta$ in Algorithm 2). Instead of sorting $(-y^i \cdot {\boldsymbol{\beta}_{\text{C},j}}^\top (\boldsymbol{z}_j^\star - \boldsymbol{z}_j^i)), j \in [m]$, for each sub-problem $\delta \in \{0, 1, \ldots, m\}$, Algorithm 2 sorts those values once in the beginning.

The discussion so far establishes the correctness of Algorithm 2. In view of its runtime, we note that the aforementioned sorting takes time $\mathcal{O}(m \log m)$ as each value $(-y^i \cdot {\boldsymbol{\beta}_{\text{C},j}}^\top (\boldsymbol{z}_j^\star - \boldsymbol{z}_j^i)), j \in [m]$ can be computed in constant time, determining $\boldsymbol{z}_j^\star$ for all $j \in [m]$ takes time $\mathcal{O}(k)$ since the variables corresponding to each feature vanish in all but one (known) location, and determining the set $\mathcal{J}$ for each $\delta \in \{0, 1, \ldots, m\}$ takes time $\mathcal{O}(n + m^2)$ since the expression $-y^i {\boldsymbol{\beta}_{\text{N}}}^\top \boldsymbol{x}^i$ only needs to be computed once. $\qquad \square$

**Proof of Theorem 6:** The statement of the theorem follows immediately from Theorems 18 and 19 of [21]. $\qquad \square$

**Proof of Lemma 1:** The assumption in the statement of the lemma allows us to choose $M \in \mathbb{R}$ such that $\mathcal{H} \subseteq [-M, +M]^{1+n+k}$. We then show that there is $\boldsymbol{\xi}^0 \in \Xi$ and $C > 0$ such that

$$l_{\boldsymbol{\beta}}(\boldsymbol{\xi}) \leq C[1 + d(\boldsymbol{\xi}, \boldsymbol{\xi}^0)] \quad \forall \boldsymbol{\beta} \in [-M, M]^{1+n+k}, \ \forall \boldsymbol{\xi} \in \Xi,$$

that is, for all $\boldsymbol{\beta} \in [-M, M]^{1+n+k}$ and all $\boldsymbol{\xi} \in \Xi$ we have

$$\log \left( 1 + \exp \left[ -y \cdot \left( \beta_0 + {\boldsymbol{\beta}_{\text{N}}}^\top \boldsymbol{x} + {\boldsymbol{\beta}_{\text{C}}}^\top \boldsymbol{z} \right) \right] \right)$$

$$\leq C \left[ 1 + \|\boldsymbol{x} - \boldsymbol{x}^0\| + \left( \sum_{i \in [m]} \mathbb{1}[\boldsymbol{z}_i \neq \boldsymbol{z}_i^0] \right)^{1/p} + \kappa \cdot \mathbb{1}[y \neq y^0] \right].$$

To see this, fix any $\boldsymbol{\xi}^0 = (\boldsymbol{0}, \boldsymbol{z}^0, y^0) \in \Xi$. Since $\kappa \cdot \mathbb{1}[y \neq y^0] \geq 0$ and $\sum_{i \in [m]} \mathbb{1}[\boldsymbol{z}_i \neq \boldsymbol{z}_i^0])^{1/p} \geq 0$ on the right-hand side of the above inequality, it suffices to show that

$$\log \left( 1 + \exp \left[ -y \cdot \left( \beta_0 + {\boldsymbol{\beta}_{\text{N}}}^\top \boldsymbol{x} + {\boldsymbol{\beta}_{\text{C}}}^\top \boldsymbol{z} \right) \right] \right) \leq C \left[ 1 + \|\boldsymbol{x}\| \right] \quad \forall \boldsymbol{\beta} \in [-M, M]^{1+n+k}, \ \forall \boldsymbol{\xi} \in \Xi.$$

Note that the left-hand side of this inequality can be bounded from above by

$$\begin{aligned}
1 + \exp \left[ -y \cdot \left( \beta_0 + {\boldsymbol{\beta}_{\text{N}}}^\top \boldsymbol{x} + {\boldsymbol{\beta}_{\text{C}}}^\top \boldsymbol{z} \right) \right] \ & \leq \ 1 + \exp \left[ \left| \beta_0 + {\boldsymbol{\beta}_{\text{N}}}^\top \boldsymbol{x} + {\boldsymbol{\beta}_{\text{C}}}^\top \boldsymbol{z} \right| \right] \\
& \leq \ 2 \exp \left[ \left| \beta_0 + {\boldsymbol{\beta}_{\text{N}}}^\top \boldsymbol{x} + {\boldsymbol{\beta}_{\text{C}}}^\top \boldsymbol{z} \right| \right].
\end{aligned}$$

It is therefore sufficient to show that

$$\log(2\exp\left[\left|\beta_0 + \boldsymbol{\beta_N}^\top \boldsymbol{x} + \boldsymbol{\beta_C}^\top \boldsymbol{z}\right|\right]) \le C[1 + \|\boldsymbol{x}\|] \qquad \forall \boldsymbol{\beta} \in [-M, M]^{1+n+k},$$
$$\forall \boldsymbol{\xi} \in \Xi$$

$$\iff \log(2) + \left|\beta_0 + \boldsymbol{\beta_N}^\top \boldsymbol{x} + \boldsymbol{\beta_C}^\top \boldsymbol{z}\right| \le C[1 + \|\boldsymbol{x}\|] \qquad \forall \boldsymbol{\beta} \in [-M, M]^{1+n+k},$$
$$\forall \boldsymbol{\xi} \in \Xi$$

$$\iff \log(2) + \max_{\boldsymbol{\beta} \in [-M, M]^{1+n+k}} \left\{\left|\beta_0 + \boldsymbol{\beta_N}^\top \boldsymbol{x} + \boldsymbol{\beta_C}^\top \boldsymbol{z}\right|\right\} \le C[1 + \|\boldsymbol{x}\|] \qquad \forall \boldsymbol{\xi} \in \Xi$$

$$\iff \log(2) + (M + M\|\boldsymbol{x}\| + Mk) \le C[1 + \|\boldsymbol{x}\|] \qquad \forall \boldsymbol{\xi} \in \Xi$$

$$\iff \log(2) + M(1+k) + M\|\boldsymbol{x}\| \le C + C\|\boldsymbol{x}\| \qquad \forall \boldsymbol{\xi} \in \Xi.$$

The last condition, however, is readily seen to be satisfied by any $C \ge \log(2) + M(1+k)$. $\qquad \square$

**Proof of Theorem 7:** The statement of the theorem follows directly from Theorem 20 of [21]. $\quad \square$

**Proof of Theorem 8:** The existence of an optimal solution follows from Theorem 3 of [42], and the sparsity of the optimal solution is due to Theorem 4 of [42]. Note that the assumptions of those theorems are satisfied since the Wasserstein ball $\mathfrak{B}_\epsilon(\widehat{\mathbb{P}}_N)$ is centered at the empirical distribution $\widehat{\mathbb{P}}_N$, which has finite moments by construction. $\qquad \square$