# OpenReview forum: "Wasserstein Logistic Regression with Mixed Features"
_NeurIPS.cc/2022/Conference — NeurIPS 2022 Accept_

### Official Review · Reviewer_tKKP · 2022-07-10

**Rating:** 6
**Confidence:** 3
**Soundness:** 3 good
**Presentation:** 4 excellent
**Contribution:** 3 good

**Summary:**

This paper addresses distributionally robust logistic regression with mixed features (i.e., continuous and categorical). Formally, the logistic regression criterion is minimized for any distribution in a Wasserstein neighborhood of the empirical distribution, see Eq. (1). The categorical nature of the features considered yields a more complex equivalent problem than in previous works [28], see Theorem 1. The authors then show the importance of considering categorical features as non continuous (Section 2.2), and that their problem cannot be rephrased as a regularized logistic regression (Theorem 3). The equivalent problem derived in Thm 1 is however NP-hard to solve for a generic loss function (Theorem 2), but the authors devise an algorithm to solve it with the log loss (Theorem 4). Experiments are also provided (Section 4).

**Questions:**

Major:
- regarding the justification for not considering categorical values as continuous: why considering the absolute value and not the square difference (possibly scaled by 1/4 if necessary)? does it make any difference? how come that $z=15,312$ is considered, this distribution cannot be in the ball centered at $\hat{P}_N$, or am I missing something here?
- Instead of saying *generic loss*, I would specify that a linear loss is used in Theorem 2
- I find Theorem 3 very interesting, and would give a brief description of the proof, as it holds for **any** loss, what I believe is a strong result
- what happens if the discrepancy between measures to create $\mathcal{B}_\epsilon$ is not the Wasserstein?

Minor:
- l.91: $\sum_j z_j$ should be equal to 1 or less or equal than 1?
- l.92: given the proposed definition, $\mathbb{C}(2)$ should be {$(0,0), (0, 1), (1, 0)$}, no?
- display 97-98: have the dimensions aligned with the $\beta$, i.e., put $\beta_0$ in the end, or have $\mathbb{R}^{1 + n + k}$, could help
- the notation $d$ for the distance in the Wasserstein definition (2) might be confusing with $d\xi$
- I found formulation (6) from the appendix more readable and understandable than Eq. (4), although I understand the desire for a convex formulation. Maybe adding a remark to explain how to make the softplus constraints convex can be a solution
- Fig. 1: add "robust" in the legend for the methods concerned (left), $(z, y) = $ on the $x$ axis no?

**Limitations:**

This is a theoretical work with no foreseeable negative societal impact.

**Strengths And Weaknesses:**

Strengths:
- the paper is clear and well written
- the authors provide a sound and thorough study of the problem

Weaknesses:
- I am not extremely familiar with the literature, but this work might be considered to be closely related to the reference [28], that already studies distributionally robust logistic regression, but with continuous features only. Though, I feel that the categorical features yields problems of a significantly different nature. Theorem 3 is a perfect illustration of this fact.
- I am not 100% convinced by the proof reasoning which shows that categorical features cannot be treated as continuous ones, see my questions below.

Overall I like the paper and tend towards acceptance. My only concern regards its significance, with respect to [28] in particular, although I feel that the problems derived are sufficiently different in nature to justify a novel publication. I am open to discussion with the authors and other reviewers familiar with the literature on this point.

---

> ### Author Response · Authors · 2022-08-02
> **Response to Reviewer tKKP - Part 1**
>
> ### Answer to Weakness \# 1
> We apologize for not making sufficiently clear our contributions in the initial paper draft. We agree that model (4) derived in Theorem 1 is closely related to the model of [28], and that this derivation is not a major contribution of our paper. Instead, the key contributions of the present paper are *(i)* a complexity analysis of this problem, which shows that the problem becomes NP-hard in general when categorical features are present ([28] only studies the polynomial-time solvable case of continuous-only features) yet remains polynomial-time solvable in many special cases of broad interest; *(ii)* a proof that in contrast to every other Wasserstein machine learning paper known to us, model (4) does not reduce to a regularized problem variant (as you pointed out in your comment as well); and *(iii)* an efficient column-and-constraint solution scheme that, as we found out after the submission of the manuscript, is very general and extends to several other machine learning problems (including linear and quantile regression, support vector machines, decision and regression trees as well as the kernel trick). Please note that we are not aware of any other Wasserstein machine learning approach that can explicitly handle categorical features. We have sharpened our presentation of the contributions in the revised manuscript and hope that they more accurately reflect our additions to the literature.
>
> ### Answer to Weakness \# 2
> We sincerely thank you for challenging us to more clearly present the advantages of handling categorical features explicitly, that is, the value that our method adds above and beyond that of [28]. The revision addresses this point in two ways.
>
> Firstly, we add more details regarding the worst-case distribution of Figure 1 to a dedicated section in the appendix. We elaborate on this further in our response to your major question 1 below.
>
> Secondly, we have significantly improved the presentation of our numerical results. We have now repeated the numerical results for the pure-categorical features setting with 100 training/test splits, and we have included *(i)* the approach of [28] that treats categorical features as continuous ones and *(ii)* the robust Wasserstein profile inference approach of Blanchet et al. (2019). Our new results show that we outperform all competitors in a statistically significant way on many of the datasets. (Note that some datasets were taken out as per the request of another reviewer.) Please note that the results for the mixed-feature case are still running, and they will be included in the camera-ready version. Since those results were already better than those of the pure-categorical case in the first draft of the paper, however, we have no doubt that our method will perform well there, too.
>
> We would like to highlight that, thanks to the reviews we received, we increased the number of training/test splits from 20 to 100 and reported mean errors. Due to these changes, we are now strictly outperforming benchmark models more often. Out of the 14 categorical datasets, our models are achieving the best error in 12 datasets and in 9 of them our improvements are statistically significant over all other benchmark models. The details of the t-tests applied to the out-of-sample errors for measuring statistical significance can be found in the updated Appendices.
>
> ### Answer to Major Question \# 1
> Thank you for this question, and we apologize for the omission in the first draft of the paper! We do not believe that replacing the absolute value with a square difference would change our findings in any significant way. Instead, the key issue is that the approach of [28] cannot accommodate for a support for categorical features. The revised version of the manuscript now points out that the only previously known tractable reformulations for Wasssertein DRO problems including support are for piecewise affine loss functions, see Theorem 14 in Section 3.2 of [i]. For more general continuous loss functions (as the log-loss employed in our paper), no tractable reformulation with support constraints appears to be known. Hence, the support of the model from [28] cannot be restricted to $[-1, 1]$, and as a result the worst-case distribution places strictly positive probability on pathological realizations. We further discuss this issue in a new appendix where we also derive the worst-case distribution of [28] in closed form.
>
> #### References
> [i] - Regularization via Mass Transportation, JMLR 20:1--68, 2019.
>
> ### Answer to Major Question \# 2
> Please note that our counterexample in the proof of Theorem 2 is *not* a linear loss-function as it contains an indicator function. It attains the value $\boldsymbol{c}^\top \boldsymbol{z}$ if $\boldsymbol{F} \boldsymbol{z} \leq \boldsymbol{g}$ and $- \mathrm{M}$ otherwise. In fact, linear loss functions could be handled in polynomial time.

---

> ### Author Response · Authors · 2022-08-02
> **Response to Reviewer tKKP - Part 2**
>
> ### Answer to Major Question \# 3
> Thank you; we are glad to hear that you find this result interesting! In the revised version of the manuscript we try to give some further insight into why this result holds; we hope you find this discussion valuable.
>
> ### Answer to Major Question \# 4
> This is an interesting point. One could in principle replace the Wasserstein ball with a $\phi$-divergence ball (such as the KL-divergence) and study the resulting DRO problem. Generally speaking, Wasserstein and $\phi$-divergences are complementary technologies and there is not "better" or "worse" approach; so either formulation would lead to a DRO problem that we would consider worthy studying. That said, a $\phi$-divergence ball centered at an empirical distribution will only contain those distributions that are absolutely continuous with respect to the reference distribution $\widehat{\mathbb{P}}_N$, and hence the worst-case distribution cannot place positive probability mass on feature combinations that have not been seen in the data. This is undesirable in our context: Imagine we have a problem with $K$ binary features and $N$ empirical samples. There are $2^K$ ways to choose those binary feature values, and typically $N \ll 2^K$. So we would exclude most of the possible feature combinations from appearing in the worst-case distribution if we were to replace the Wasserstein ball with a KL-divergence ball, say. This problem also exists in the continuous case, but it is particularly prevalent in the categorical and mixed-feature cases. We also refer to the paper [i] which shows via an example that if we use $\phi$-divergence, we do not get a meaningful ambiguity ball around the empirical distribution. That is in order to include the real data-generating distribution, we have no choice but to also consider very weird pathological distributions.
>
> #### References
> [i] - Distributionally Robust Stochastic Optimization with Wasserstein Distance, arXiv:1604.02199, 2016.
>
> ### Answer to Minor Comments
> Thank you for your careful reading of the manuscript! In response to your comments:
>
> 1. It should indeed be less than or equal to 1, as opposed to equal to 1, as we are using a one-hot encoding where the last category is dropped.
> 2. For the same reason, we believe $\mathbb{C}(2) = \mathbb{B}$ is indeed correct.
> 3. Thank you for this suggestion; we have implemented this as you suggested!
> 4. Thank you; we agree that the difference between the italicized and roman 'd's is difficult to spot. We now alert the reader to this difference in the revised version of the notation section.
> 5. Thank you! We agree that formulation (6) is easier to parse. We have nevertheless kept formulation (4) as our column-and-constraint generation scheme is built around it. That said, we have now added a remark after the theorem which explains how formulation (4) emerges from the more readily accessible formulation (6) that can be found in the appendix. We hope that this aids the exposition.
> 6. Thank you; we have implemented your suggestion!

---

> ### Comment · Reviewer_tKKP · 2022-08-06
> **Acknowledging feedback**
>
> I thank the authors for their response. I find the remark on $\phi$-divergence interesting, and an additional motivation for the studied framework. Indeed you are right about the loss. Still my point was to replace "for generic loss" by "there exists a loss", as the current statement may suggest that the problem is NP-hard for _any loss_.
>
> I have read the other reviews and responses by the authors. Overall my stance on the paper did not change, and I still think that the paper's pros outweigh its limitations. Hence I keep my score.

---

> > ### Author Response · Authors · 2022-08-07
> > **We thank Reviewer tKKP**
> >
> > We thank you again for your comments! We agree with you on the statement of Theorem 2 and will clarify that in our next revision of the paper.

---

### Official Review · Reviewer_53YQ · 2022-07-11

**Rating:** 6
**Confidence:** 3
**Soundness:** 3 good
**Presentation:** 3 good
**Contribution:** 3 good

**Summary:**

To overcome the non-robustness in classical logistic regression, the authors consider the distributionally robust logistic regression by introducing an ambiguity set defined by the popular Wasserstein distance.
In contrast to the previous literature, the proposed method focus on mixed features. Despite the optimization problem itself being exponential-size for generic losses, the authors show that it can be solved in polynomial time using the column-and-constraint generation scheme.
Theoretically, they provide finite-sample and asymptotic performance guarantees for the proposed method.
Experiments on synthetic and real-world datasets show the accuracy and efficiency of the proposed method in both unregularized and regularized versions.

**Questions:**

**Major Comments**

* My major concerns are about the experiments in Section 4.
1. Sections 4.2--4.3: I do not find the proposed DRO (resp. r-DRO) performs much better than LR (resp. r-LR), while DRO (resp. r-DRO) might require a longer runtime than LR (resp. r-LR). The authors should provide comparison results on both performance and time to show the accuracy-time tradeoff.
2. As the authors discussed in Section 1, the motivation to consider the robust version of LR is that classical LR is prone to overfitting, especially in face of outliers or distribution shifts. However, this kind of setting has not been considered in experiments. The authors should provide simulations or real data analysis on these specific settings to show the advantages of DRO (resp. r-DRO) over classical LR (resp. r-LR).
3. Sections 4.2--4.3: Only classical logistic regression (with/without regularization) is compared with the proposed method. I suggest the authors provide more comparisons with existing distributionally robust regression methods (e.g., "Distributionally
robust logistic regression" and "Robust Wasserstein profile inference and
applications to machine learning." ) w.r.t. both computational time and classification error.
4. The authors take the medians over 20 random training set-test set splits as the final results in Tables 1--2. What if the results are taken as means (rather than medians)? Does the proposed method still outperform the competitors?
5. For some of the UCI datasets in Table 1 (e.g., lenses with $N=24$ and balloons with $N=16$), the sample size $N$ is too small to provide a convincing classification error on the test set.

* Wasserstein distance estimation usually suffers from the curse-of-dimensionality, see "Projection-based techniques for high-dimensional optimal transport problems". In particular, the convergence rate of the empirical Wasserstein distance may be arbitrarily slow as the size of dimensions increases. I wonder how does the number of variables affect the performance of the proposed method? Some discussion and empirical results are needed.

**Minor Comments**

1. The definition of $\mathbb{B}$ seems to be inconsistent in lines 16 and line 73.
2. In problem (4), the notations need to be defined and explained in detail.
3. In Section 3, please add some references for the column-and-constraint generation scheme.
4. In Section 4.1, the comparison between the left panel and the middle panel of Fig. 2 is unfair for their scales are not consistent. Please provide the log-scale runtime versus increasing $m$.

**Limitations:**



**Strengths And Weaknesses:**

In general, the paper is well-written and easy to follow, the motivation to consider categorical variables is clearly stated, and the theoretical results also seem to be correct.
As far as I have read, the work provides novel discussions and results which are not captured by the cited literature.
However, I have some concerns about the numerical experiments and also some suggested corrections/additions, which are presented below.

---

> ### Author Response · Authors · 2022-08-02
> **Response to Reviewer 53YQ - Part 1**
>
> ### Answer to Question / Major Comment  1
> We sincerely thank you for challenging us to improve the presentation of our numerical results. We have now repeated the numerical results for the pure-categorical features setting with 100 training/test splits, and we have included *(i)* the approach of [28] that treats categorical features as continuous ones and *(ii)* the robust Wasserstein profile inference approach of Blanchet et al. (2019). Our new results show that we outperform all competitors in a statistically significant way on many of the datasets. (Note that some datasets were taken out as per your major comment 5.) Please note that the results for the mixed-feature case are still running, and they will be included in the camera-ready version. Since those results were already better than those of the pure-categorical case in the first draft of the paper, however, we have no doubt that our method will perform well there, too. Moreover, as you requested we now also include the computation times for all methods on all instances. We agree that there is a trade-off between performance and computation times, and the specific circumstances will determine which method is most appropriate. That said, we believe that our revised numerical results show that our proposed approach lies on the "efficient performance-runtime frontier."
>
> ### Answer to Question / Major Comment  2
> Thank you for this clarification question! We believe we did not state that distributionally robust optimization alleviates the issue of outliers, but that it can alleviate the issues of overfitting, erroneous feature and label values as well as distribution shifts. These issues have been investigated in some detail in the related literature, and due to a lack of space and time the revised manuscript now refers the interested readers to the publications [i, ii] for overfitting, the reference [ii] for label uncertainty and the references [iii, iv] for distribution shifts, respectively. We hope that this satisfactorily addresses your comment.
>
> #### References
> [i] - Data-driven distributionally robust optimization using the Wasserstein metric: Performance guarantees and tractable reformulations, Mathematical Programming 171:115--166, 2018.
>
> [ii] - Wasserstein Distributionally Robust Optimization: Theory and Applications in Machine Learning, INFORMS TutORials in Operations Research, 2019.
>
> [iii] - Sequential Domain Adaptation by Synthesizing Distributionally Robust Experts, PMLR 139:10162--10172, 2021.
>
> [iv] - Robust Generalization despite Distribution Shift via Minimum Discriminating Information, NeurIPS 2021.
>
> ### Answer to Question / Major Comment  3
> Thank you for this suggestion! As we elaborated in our response to your first major comment, we now compare our method against both of these approaches as well in the revised manuscript.
>
> ### Answer to Question / Major Comment  4
> Thank you. In the initial draft, we considered the medians as we only conducted 20 training-test splits. Since we now conduct 100 training-test splits, we used the mean. Due to these changes, we are now strictly outperforming benchmark models more often. Out of the 14 categorical datasets, our models are achieving the best error in 12 datasets and in 9 of them our improvements are statistically significant over all other benchmark models. The details of the t-tests applied to the out-of-sample errors for measuring statistical significance can be found in the updated Appendices.
>
> ### Answer to Question / Major Comment  5
> Thank you. We agree, and we have removed those datasets from the paper.
>
> ### Answer to Question / Major Comment  6
> This is a very sharp comment. Our performance guarantees in Appendix A scale with the dimension of the feature space as we seek for a high confidence of the unknown true distribution being contained in our ambiguity set $\mathfrak{B}_\epsilon (\widehat{\mathbb{P}}_N)$. A dimension-independent performance guarantee can be obtained along the lines of [i] if one instead only seeks for a high confidence of the unknown true model $\boldsymbol{\beta}^\star$ being contained in the union of argmax's corresponding to the individual distributions $\mathbb{P}$ contained in $\mathfrak{B}_\epsilon (\widehat{\mathbb{P}}_N)$. We added a discussion of this point, together with a pointer to the review paper you have provided, in Appendix A. Please note that these performance guarantees are in some sense orthogonal to the developments of our paper, and for this reason, we only mention them for completeness' sake in the appendix (rather than the main paper).
>
> On the empirical side, we observe that our mixed-feature Wasserstein DRO approach appears to work well across the UCI data sets that we considered, which include data sets with low-dimensional as well as higher-dimensional feature spaces.
>
> #### References
> [i] - Robust Wasserstein Profile Inference and Applications to Machine Learning, Journal of Applied Probability 56(3):830--857, 2019.

---

> > ### Comment · Reviewer_53YQ · 2022-08-09
> > **response to rebuttal**
> >
> > Thank you for your detailed explanations. I have no more questions and would tend to raise my rating from 5 to 6 (weak accept).

---

> ### Author Response · Authors · 2022-08-02
> **Response to Reviewer 53YQ - Part 2**
>
> ### Answer to Minor comments
> Thank you; we have addressed all of these comments as you suggested.

---

### Official Review · Reviewer_ZDos · 2022-07-11

**Rating:** 7
**Confidence:** 4
**Soundness:** 3 good
**Presentation:** 3 good
**Contribution:** 3 good

**Summary:**

This paper studies the distributionally robust logistic regression problem with mixed (numerical and categorical) features. The problem is proved to be solvable by a polynomial-time solution scheme. In particular, this paper develops a column-and-constraint approach to solve the problem. This paper also shows that the distributionally robust logistic regression problem is not equivalent to regularized logistic regression as in previous works with numerical features only. When categorical features are present, this paper proves that the problem is not equivalent to regularized problems for any regularizer.

**Questions:**

I have a question regarding the example shown in Figure 1. By treating the categorical features as continuous ones, the authors claimed that the worst-case distribution places non-zero probability on an extreme scenario 15, 312, I am wondering would such scenarios be avoided when we restrict the range of the continuous features, like within [-1,1]. Also, it would be better if the author could comment on some theoretical comparisons between the performance of the Mixed-feature model method and the continuous-feature model in sec 2.2.
Typos:
1. In algorithm 2, delta \in \{0,1,2,...\}, a typo in the first comma.

**Limitations:**

It seems that the authors do not mention limitations of the proposed method.

**Strengths And Weaknesses:**

Strengths

1. This paper studies distributionally robust logistic regression with mixed features, while previous works mainly focus on numerical features only or simply treat the categorical feature as numerical ones.
2. This paper proposes a column-and-constraint approach to solve the problem efficiently.
3. This paper shows that the problem with mixed features is not equivalent to regularized problems, as is commonly the case for distributionally robust logistic regression with numerical features.
4. Numerical results on multiple real datasets are provided.

Weaknesses
1. The main advantage and motivation of not dealing with the categorical features directly are only shown in the example of a worst-case distribution in Figure 1, which might be insufficient and the results could be stronger if the author could have certain theoretical comparisons.

---

> ### Author Response · Authors · 2022-08-02
> **Response to Reviewer ZDos**
>
> ### Answer to Weakness # 1
> We sincerely thank you for challenging us to more clearly present the advantages of handling categorical features explicitly, that is, the value that our method adds above and beyond that of [28]. The revision addresses this point in two ways.
>
> Firstly, we add more details regarding the worst-case distribution of Figure 1 to a dedicated section in the appendix. In that section, we derive the worst-case distribution theoretically in closed-form. We hope that this addresses what you referred to when you asked for "theoretical comparisons."
>
> Secondly, we have significantly improved the presentation of our numerical results. We have now repeated the numerical results for the pure-categorical features setting with 100 training/test splits, and we have included *(i)* the approach of [28] that treats categorical features as continuous ones and *(ii)* the robust Wasserstein profile inference approach of Blanchet et al. (2019). Our new results show that we outperform all competitors in a statistically significant way on many of the datasets. (Note that some datasets were taken out as per the request of another reviewer.) Please note that the results for the mixed-feature case are still running, and they will be included in the camera-ready version. Since those results were already better than those of the pure-categorical case in the first draft of the paper, however, we have no doubt that our method will perform well there, too.
>
> We would like to highlight that, thanks to the reviews we received, we increased the number of training/test splits from 20 to 100 and reported mean errors. Due to these changes, we are now strictly outperforming benchmark models more often. Out of the 14 categorical datasets, our models are achieving the best error in 12 datasets and in 9 of them our improvements are statistically significant over all other benchmark models. The details of the t-tests applied to the out-of-sample errors for measuring statistical significance can be found in the updated Appendices.
>
> ### Answer to the Question
> Thank you for this question, and we apologize for the omission in the first draft of the paper! The revised version of the manuscript now points out that the only previously known tractable reformulations for Wasssertein DRO problems including support are for piecewise affine loss functions, see Theorem 14 in Section 3.2 of [i]. For more general continuous loss functions (as the log-loss employed in our paper), no tractable reformulation with support constraints appears to be known. Hence, the support of the model from [28] cannot be restricted to $[-1, 1]$ as suggested.
>
> #### References
> [i] -  Regularization via Mass Transportation, JMLR 20:1--68, 2019.
>
> ### Answer to the Typo
> Thank you very much for pointing out the typo; this is now corrected in the revised manuscript.
>
> ### Answer to the Limitation
> We apologize if we misunderstood the new guidelines regarding the limitations; we had discussed them in the attached checklist but did not include them in the main paper. The revised numerical results now clearly point out the limitation that our approach does not always outperform the standard logistic regression. We cannot think of any other major limitations, as we are solving a variant of a well-established problem whose advantages and shortcomings are by now relatively well understood. We hope that this satisfactorily addresses your comment.

---

### Official Review · Reviewer_7VSn · 2022-07-11

**Rating:** 4
**Confidence:** 4
**Soundness:** 3 good
**Presentation:** 4 excellent
**Contribution:** 2 fair

**Summary:**

This paper considers the mixed feature distributionally robust logistic regression problem, which extends the setting in [1] to the mixed feature (i.e., the feature space can be divided into the numerical (real number) and categorical (integer) parts.  The main contribution of this paper lies in deriving an equivalent dual reformulation of this new model and further developing tractable methods to address it. More specifically, by fully exploiting the problem-specific structure arising from the log-loss, the authors fully utilize its exponential conic representation and provide a Column-and-Constraint Solution Scheme to solve it. Finite steps convergence guarantee has also been established. Extensive experiments have been shown to demonstrate the effectiveness of its model performance.

[1]: Shafieezadeh-Abadeh, Soroosh et al. “Distributionally Robust Logistic Regression.” NIPS (2015).


**Questions:**

No

**Ethics Review Area:**

["I don’t know"]

**Limitations:**

The computational paradigm and theoretical results (i.e., polynomial solvable）is restricted to the logistic regression case.

**Strengths And Weaknesses:**

**Strengths**:

The mixed features considered in the paper are less explored in distributionally robust optimization. This paper takes a first step towards filling this gap. The paper is easy to follow and well-organized. Comprehensive experiments have been given to illustrate the extension.  Notably, Theorem 3 seems very interesting to me. They point out the absence of reformulation as a regularized problem, while this property holds for many DRO problems.

**Weakness**:

There are three major concerns:

1. The novelty of both modeling and techniques is moderate. The proposed model is a natural extension based on [1]. The dual reformulation is also relatively standard if we adopted the method provided in [1].

2. All methodologies developed in this paper highly depend on the log-loss and specific cost function (see theorem 2 for details), which are rather restrictive. In my opinion, the impact of this paper on broad ML tasks is limited.

3. On the computational side, although the paper gives a polynomial-time solvable method to address the resulting dual reformulation, it still relies on certain off-the-shelf solvers, which will potentially suffer from the scalability issue on medium or large-scale datasets. More importantly, for the experiment results, the proposed mixed feature model does not gain substantial improvements over other baselines (i.e., vanilla logistic regression or distributionally robust LR) in terms of test accuracy (i..e, see tables 1 and 2),  but instead cost too much computational effort.

For the above reasons, this paper is strictly below the neurips acceptance level.

---

> ### Author Response · Authors · 2022-08-02
> **Response to Reviewer 7VSn**
>
> ### Answer to Weakness \# 1
> We apologize for not making sufficiently clear our contributions in the initial paper draft. We agree that model (4) derived in Theorem 1 is a more or less straightforward extension of [28]. Instead, the key contributions of the present paper are *(i)* a complexity analysis of this problem, which shows that the problem becomes NP-hard in general when categorical features are present ([28] only studies the polynomial-time solvable case of continuous-only features); *(ii)* a proof that in contrast to every other Wasserstein machine learning paper known to us, model (4) does not reduce to a regularized problem variant; and *(iii)* an efficient column-and-constraint solution scheme. Please note that we are not aware of any other Wasserstein machine learning approach that can explicitly handle categorical features. We have sharpened our presentation of the contributions in the revised manuscript and hope that they more accurately reflect our additions to the literature.
>
> ### Answer to Weakness \# 2
> Thank you for raising this important point! In fact, we found out after the submission of the manuscript that our column-and-constraint generation scheme is very general and extends to several other machine learning problems, including linear and quantile regression, support vector machines, decision and regression trees as well as the kernel trick. This is in some sense reminiscent of the generalization of [28] to other machine learning problems in [i]. While the size limitations of the NeurIPS proceedings do not allow us to explore this generalization in the present paper, we have added this point in the conclusions of the revised manuscript.
>
> #### References
> [i] - Regularization via Mass Transportation, JMLR 20:1--68, 2019.
>
> ### Answer to Weakness \# 3.1 (Reliance on Off-the-Shelf Solvers)
> While our solution approach does indeed utilize the MOSEK solver, we would like to point out that the software is free for academic use and that there are several powerful open-source solvers available to solve exponential cone programs, such as Ipopt and CVXOpt. Our approach does not rely on any particular feature of MOSEK, and our source code (which will be made open-source upon acceptance, and is currently withheld only to maintain the anonymity of the review process) can readily be adapted to work with these open-source solvers. Also, we see our present paper as a first step in the direction of developing custom algorithms for this problem. This is similar to the original optimization paper [23] on Wasserstein problems, which has relied on standard solvers and which has subsequently spurred developments of tailored first-order solution methods (e.g. [i-iii]). The introduction to Section 4 and the conclusion of the revised manuscript now clarify those points; thank you!
>
> #### References
> [i] -  Stochastic Optimization for Regularized Wasserstein Estimators, PMLR 119:602-612, 2020.
>
> [ii] - First-Order Methods for Wasserstein Distributionally Robust MDP, PMLR 139:2010--2019, 2021.
>
> [iii] - Stochastic Gradient Descent in Wasserstein Space, arXiv:2201.04232, 2022.
>
> ### Answer to Weakness \# 3.2 (Lack of Substantial Improvements)
> We sincerely thank you for challenging us to improve the presentation of our numerical results. We have now repeated the numerical results for the pure-categorical features setting with 100 training/test splits, and we have included *(i)* the approach of [28] that treats categorical features as continuous ones and *(ii)* the robust Wasserstein profile inference approach of Blanchet et al. (2019). Our new results show that we outperform all competitors in a statistically significant way on many of the datasets. (Note that some datasets were taken out as per the request of another reviewer.) Please note that the results for the mixed-feature case are still running, and they will be included in the camera-ready version. Since those results were already better than those of the pure-categorical case in the first draft of the paper, however, we have no doubt that our method will perform well there, too.
>
> We would like to highlight that, thanks to the reviews we received, we increased the number of training/test splits from 20 to 100 and reported mean errors. Due to these changes, we are now strictly outperforming benchmark models more often. Out of the 14 categorical datasets, our models are achieving the best error in 12 datasets and in 9 of them our improvements are statistically significant over all other benchmark models. The details of the t-tests applied to the out-of-sample errors for measuring statistical significance can be found in the updated Appendices.
>
> ### Answer to Limitations
> Please refer to our earlier responses.

---

### Official Review · Reviewer_Qih9 · 2022-07-12

**Rating:** 6
**Confidence:** 4
**Soundness:** 3 good
**Presentation:** 3 good
**Contribution:** 2 fair

**Summary:**

This paper proposes a distributionally robust logistic regression with mixed-type features.
Optimization method is considered. Some properties related to complexity analysis were given.
A number of experiments were performed on classical datasets.


**Questions:**

The presentation is  clear. The discussions are pretty clear.
The strengths and weaknesses can be found in above comments.
No questions are raised here.



**Limitations:**

Yes

**Strengths And Weaknesses:**

Strengths:
1.	It considers the expansion of recent distributionally robust logistic regression (LR) with continuous features to that with mixed features.
2.	The complexity analysis is given. In particular, it proves the the non-existence of regularized version for the formulation derived from Wasserstein distance.

Weaknesses:
1.	It appears to be an incremental derivation from existing distributionally robust logistic regression (LR) with continuous features.
2.	In terms of experimental results, the proposed formulation does not appear much better than the classical LR in terms of accuracy. Actually, it is not clear there is any significant advantages when compared to LR (Please see next comment). Without significant improvement, the new formulation is more like a conceptual exercise.

3.	The comparative evaluation in lines 282-287 is not so convincing. Currently, only the number of best performance is counted;

“The table shows that for the unregularized model, both of our distributionally robust logistic
regression achieve the lowest classification error in 78% of the instances, whereas the classical logistic
regression achieves the lowest classification error in 61% of the instances. For the regularized model,
the results remain at 78% (our model) vs. 61% (classical logistic regression). Overall, the non-robust
models failed to achieve the lowest classification error in three of the instances, whereas the robust
287 models missed the lowest classification error in only one of the instances.”

This is analogous to ‘bean counting”.
However, the variance is not considered.
It would be desirable to show variance  information. Maybe some statistical test can be used for judge if there is any significant difference between the LR with the DROs, or between the regularized r-LR with the regularized r-DROs.

---

> ### Author Response · Authors · 2022-08-02
> **Response to Reviewer Qih9**
>
> ### Answer to Weakness \# 1
> We apologize for not making sufficiently clear our contributions in the initial paper draft. We agree that model (4) derived in Theorem 1 is a more or less straightforward extension of [28]. Instead, the key contributions of the present paper are *(i)* a complexity analysis of this problem, which shows that the problem becomes NP-hard in general when categorical features are present ([28] only studies the polynomial-time solvable case of continuous-only features) yet remains polynomial-time solvable in many special cases of broad interest; *(ii)* a proof that in contrast to every other Wasserstein machine learning paper known to us, model (4) does not reduce to a regularized problem variant; and *(iii)* an efficient column-and-constraint solution scheme that, as we found out after the submission of the manuscript, is very general and extends to several other machine learning problems (including linear and quantile regression, support vector machines, decision and regression trees as well as the kernel trick). Please note that we are not aware of any other Wasserstein machine learning approach that can explicitly handle categorical features. And the absence of a regularised problem variant, which appears to exist for all other formulations known to us, confirms that our formulation is indeed fundamentally different. We have sharpened our presentation of the contributions in the revised manuscript and hope that they more accurately reflect our additions to the literature.
>
> ### Answer to Weaknesses \# 2 \& \# 3
> We sincerely thank you for challenging us to improve the presentation of our numerical results. We have now repeated the numerical results for the pure-categorical features setting with 100 training/test splits, and we have included *(i)* the approach of [28] that treats categorical features as continuous ones and *(ii)* the robust Wasserstein profile inference approach of Blanchet et al. (2019). Our new results show that we outperform all competitors in a statistically significant way on many of the datasets. (Note that some datasets were taken out as per the request of another reviewer.) Please note that the results for the mixed-feature case are still running, and they will be included in the camera-ready version. Since those results were already better than those of the pure-categorical case in the first draft of the paper, however, we have no doubt that our method will perform well there, too.
>
> We would like to highlight that, thanks to the reviews we received, we increased the number of training/test splits from 20 to 100 and reported mean errors. Due to these changes, we are now strictly outperforming benchmark models more often. Out of the 14 categorical datasets, our models are achieving the best error in 12 datasets and in 9 of them our improvements are statistically significant over all other benchmark models. The details of the t-tests applied to the out-of-sample errors for measuring statistical significance can be found in the updated Appendices.

---

### Author Response · Authors · 2022-08-02
**Responses Posted & Thanks to Reviewers**

We would like to thank the reviewers for their constructive reviews. Their comments led us significantly improve our paper in terms of both theory and computation. We have replied to each reviewer and addressed every comment and question.

In the revised manuscript, we highlight the changes and new material with blue-colored text.

The revised version of our submission as well as the supplementary material will be uploaded by 3 August 2022, 9 pm UTC.

We will be active and responsive on OpenReview during the Author-Reviewer Discussion period, and we are looking forward to discussing our work further.

Thank you for your time.

---

### Meta-Review · Area_Chair_zikg · 2022-08-20

**Recommendation:** Accept
**Confidence:** Certain

**Metareview:**

The focus of the submission is distributionally robust logistic regression when the discrepancy used in the ambiguity set is the Wasserstein distance and the features are mixed (i.e., they can contain both numerical and categorical variables). After showing that the resulting optimization problem (1) with the log-loss function can be reformulated as a finite-dimensional exponential conic program (Theorem 1), they (i) prove that (1) can be solved in polynomial time (Theorem 2), (ii) show that it does not admit a regularized logistic regression form (Theorem 3) as it is the case for purely numerical features, (iii) propose a column-and-constraint solver (Theorem 4-5). The practical efficiency of the proposed method is illustrated on 14 UCI benchmarks.

Logistic regression (LR) is among the most popular tools in machine learning and statistics. Handling mixed features for LR in the distributionally robust case is a relevant problem. The submission represents a solid work combining both important theoretical and empirical insights as it was evaluated by the reviewers.

**Award:**

No

---

### Decision · Program_Chairs · 2022-09-14

Accept